

# Growth in mid-monsoon dry phases over Indian region:

# Prevailing influence of anthropogenic aerosols

Rohit Chakraborty[1], Bijay Kumar Guha[2], Shamitaksha Talukdar[*1], Madineni Venkat Ratnam[1], Animesh Maitra[3]

[1]National Atmospheric Research Laboratory, Gadanki, India,

[2]National Institute of Technology, Rourkela, India

[3]Institute of Radiophysics and Electronics, Kolkata, India

rohitc744@gmail.com, bijayguha74@gmail.com, shamit@narl.gov.in[*]

**Abstract**

A detailed investigation on the potentially drought prone regions over India has been presented in this study based on the balance between precipitation and potential evapotranspiration (PET) during the South West Asian mid-monsoon season. We methodically introduce a parameter named dry day frequency (DDF) which is found suitable to present the drought index (DI) in mid-monsoon season hence strongly associated with the possibility of drought occurrences. The present study investigates the probable aspects which influence the DDF over these regions revealing that the abundance of anthropogenic aerosols especially over urbanized location have prevailing role on the growth of DDF during last few decades. The prominent increasing trend in DDF over Lucknow (26.84° N, 80.94° E ), a densely populated urban location situated in the Indo-Gangetic plain, strongly reflects the dominant association of man-made aerosols with the increasing dry phase occurrences. Increase in DDF (~90%) during the last 60 years is observed over this urban area compared to a broader region in its surroundings.  In addition, periodic impacts of synoptic scale phenomena like ENSO (El Niño–Southern Oscillation) or SSN (Sun spot number) become weaker when the study location is downscaled towards an urbanized region. However, there still remains some unclear role of air-mass transport on DDF over the potential drought prone region of north-west India. Finally, when long term projections of DDF are drawn using the high urbanization scenario of RCP 8.5 a huge rise in dry days are seen during mid-July to mid-September (reaching up to 50 dry days by the year 2100 over Lucknow) which will be a very crucial concern for policy makers in future.



## 1. Introduction

Drought is a natural and recurrent phenomenon which occurs in all forms of climate.
Although similar to aridity in many ways, droughts are mainly temporary in nature thus it
should not be confused with the water scarcity due to excess of water demand over available
supply. On the other hand these weather extremes are more reasonably linked with the
distribution and frequency of rainfall over any region. Although, there are no generally
accepted definitions for drought (Wilhite and Glantz, 1985), the American Meteorological
Society has categorized it into four types namely: meteorological or climatological,
agricultural, hydrological and socioeconomic (Heim, 2002). A prolonged drought lasts
several months or even years while the absence or reduction of precipitation creates
meteorological droughts. On the other hand, short-term (few weeks) dryness in surface layer
could results an agricultural drought (Heim, 2002). However, when prolonged meteorological
droughts reduce the ground water level severely then hydrological droughts occur. Finally, all
first three droughts with a deficit in water availability are named as socioeconomic drought.
Among these four, the agricultural drought might be a serious issue when the farming or crop
producing in humid or sub humid zones are concerned. The situation has however become
more serious in the present due to rapid population growth across all continents, thereby also
producing a hike in their global demand (Sivakumar, 2011). Now, India is a country where
agriculture and its allied activities act as major source of livelihood and hence it is expected
to be deeply affected by drought occurrences especially if it occurs in the mid-monsoon
period (as it experiences ~80% of the annual rainfall due to the southwest monsoon).
Generally drought events originate from the deficiency in precipitation, and water
shortage over a particular region and time (Wilhite and Glantz 1985). As rainfall observation
data is available from past two centuries, mostly all the calculations of drought indices
includes this variable either single headedly or in combination with other meteorological
parameters (WMO, 1975; Tannehill, 1947). Some early drought index were simply
represented the drought duration or intensity upon satisfying the drought defining criteria,
e.g. Munger (1916) defined the drought index as the length of period without 24 hours
precipitation with a minimum of 1.27 mm. Similarly, Kincer (1919) used 30 or more
consecutive days with less than 6.35 mm daily rainfall for the process of drought
identification. Marcovitch (1930) used temperature data along with the precipitation while
Blumenstock (1942) used the length of drought in days, where the count was terminated upon
occurrence of 2.54 mm of rainfall over a span of 48 hours. Likewise, many other drought



index can be found in the past literature where precipitation has been used as a primary factor
(Bates, 1935; Palmer, 1965, 1968; Gibbs and Maher, 1967; Frere and Popov, 1979; Bhalme
and Mooley, 1980; Petrasovits, 1990; Rao et al., 1981; Heddinghaus, 1991; Tate et al., 2000;
Lloyd-Hughes and Saunders, 2002). Recently, the multi-scaler drought index like
Standardized Precipitation Index (McKee et al., 1993) is widely used by several researchers
in analysing the drought characteristics. However, no single index has the ability to precisely
represent the drought duration and intensity and its possible impacts (Wilhite and Glantz,
1985). Again, apart from the rainfall, there are also some other parameters that affects the
drought severity, e.g. potential evapotranspiration (PET) and soil water holding capacity (Dai
et al., 2004).The Palmer Drought Severity Index (Palmer, 1965) is an effective parameter
which uses all these three parameters; however, it has some limitations when applying over
climatic zones like India (Niranjan et al., 2013). In addition, gathering all these parameters in
gridded form and then quantifying the drought index will be very difficult over the Indian
region. On the other hand, the standardized precipitation–evapotranspiration index (SPEI)
uses only precipitation and temperature, and is considered to be better for analysing drought
occurrence (Begueria et al., 2010; Vicente-Serrano et al., 2010a, 2010b; Das et al., 2016).
India happens to be one of the most vulnerable drought-prone countries, as severe
droughts occur at least once in a three year time span since the past few decades (Mishra and
Singh, 2010). In addition, there are numerous instances of severe drought conditions during
Monsoon as reported in recent past (Pai and Sreejith, 2010). Consequently, several studies
have been carried out in the recent years in order to understand the drought occurrences
during the Indian summer monsoon period (Ramdas, 1950; Banerji and Chabra, 1964;
Chowdhury et al., 1989; Appa Rao, 1991; Gore and Sinha Ray, 2002). Bhalme and Mooley
(1980) defined the Drought Area Index for drought intensity assessment using monthly
rainfall distribution. Raman and Rao (1981) suggested a possible relation between summer
droughts and prolonged brake phase of southwest monsoon over the Indian sub-continent.
Parthasarathy et al. (1987) identified the extreme drought years by analysing the decade long
anomalies in the Indian summer monsoon rainfall. Tyalagadi et al. (2015) analysed more than
100 years of rainfall and identified 21 drought years, half of which were associated with El
Niño. Gadgil et al. (2003) explained the excess rainfall or drought in terms of Equatorial
Indian Ocean Oscillation (EQUINOO) during 1972 – 2002, especially during monsoon
season. Francis and Gadgil (2010) also suggested the role of El Niño Southern Oscillation
(ENSO) and EQUINOO behind the 48% deficit of June rainfall over India, although there are
contradictions behind this theory (Neena et al., 2011). Apart from these oscillations like



ENSO or IOD (Indian Ocean Dipole) there are also lots of other parameters which may have
prominent influences on drought occurrence, e.g. Himalayan ice cover, Eurasian snow cover,
the passage of intra-seasonal waves, effects of accumulated pollution etc., e.g. Krishnamurti
et al. (2010) reported the intrusion of desert air mass to be responsible towards the drought
occurrences over the central Indian region.
In general, most of the previous studies on monsoon droughts are discussed on the basis
of rainfall accumulation, and there are very few, which quantify its relation with the direct or
indirect radiative effects of aerosols (Atwater, 1970; Ensor et al., 1971; Twomey, 1977;
Albrecht, 1989; Charlson et al., 1992) while considering both rainfall and PET. Absorbing
aerosols such as black carbon (BC) or dust have the capabilities of atmospheric heating by
absorbing solar radiation, while non-absorbing aerosols (e.g. sulphates) scatter the solar
radiation have less effect over the same (Lau and Kim, 2006). Additionally, they have the
capability of modulating the cloud characteristics by altering cloud radiative properties (Li et
al., 2010; Gu et al., 2012; Dipu et al., 2013; Wencai et al., 2015). Previous studies have
shown the presence of the aerosols (mainly dust and BC), and their  ability to impact the
rainfall (depending upon their sizes) during Indian summer monsoon as described by elevated
heat pump hypothesis (Lau and Kim, 2006; Manoj et al., 2011; Vinoj et al., 2014; Das et al.,
2015; Solmon et al., 2015). During late pre-monsoon or early monsoon season, the aerosol
loading over India is nearly three times higher than the average due to the dust abundance,
which is partly dependent upon the winds, precipitation and surface temperature (Dey, 2004;
Grini and Zender, 2004;Deyand Girolamo, 2010; Wang et al., 2015; Parajuli et al., 2016).
However, the vice versa can also be true (e.g. Moorthy et al., 2007; Lau and Kim, 2006).
Very recently some new attempts were also undertaken to study the long and short term
implications of both natural and anthropogenic components in producing a hindrance to
convective rainfall especially over urbanized coastal locations which may also lead to
subsequent drought occurrences (Chakraborty et al., 2016, 2017a, 2017b, Guha et al., 2017
and Talukdar et al., 2018). Keeping all these assertions in mind, the present study has put an
effort in establishing a possible relationship between aerosol loading and summer monsoon
rainfall, consequently, over drought occurrences during this period in past few decades.
Hence a detailed investigation is presented to study the evolution of dry phase leading to
drought conditions during mid-monsoon over three Indian regions based on the balance
between precipitation and PET during the monsoon season. Next, a new parameter called dry
day frequency is used to understand the trends of drought potential over the mentioned Indian
regions. This is followed by a three pronged investigation to identify the most dominant




factor behind these trends after which future projections of DDF is observed and explained
for these locations during the mid-monsoon period.

**2.   Dataset and methodology**
Most of the research attempts in recent past have employed SPE) as an indicator of drought
occurrence over the Indian region (Beguería et al., 2010; Vicente et al., 2010b; Das et al.,
2016). SPEI which is precipitation minus PET mainly represents the climatic monthly water
budget. Interestingly, this parameter is found to be the most reliable identifier of drought
occurrences as it can be expressed in terms of standardized Gaussian variance with zero mean
and one standard deviation (Vicente et al., 2010b). Another advantage of using SPEI over any
other multi-scalar drought indicators (e.g. SPI) is that it not only includes the effect of the
evaporative demand in its calculation, but also can be calculated for different time scales
(Beguería et al., 2010), unlike the PDSI which rely on a water balance of a particular system.
In this study the SPEI is calculated using monthly precipitation and PET from the CRU TS3
dataset (http://badc.erc.ac.uk/data/cru/), where the PET is calculated considering the monthly
mean temperature and the geographical location of the concerned region as per the method
suggested by Thornthwaite (1948). Hence, it provides long-term information about the
drought conditions over any location with a high spatial resolution of 0.5°×0.5° at monthly
basis. However, the available precipitation ($P$) data is provided in form of monthly
accumulated value, whereas, the $PET$ represents the monthly mean. Therefore, the difference
($D$) or SPEI is calculated for each month as follows:

$$D = P - (PET \times \text{number of days in a month}) \qquad (1)$$

It may be noted that for this analysis, this value of D is normalized with respect to the
climatic mean and 1 sigma standard deviation to obtain comparable values for all regions of
the country. These normalized values of $D$ are hereafter referred to as DI.  This study
considered the length of the dry phase as an indicator of drought occurrence and severity,
which is calculated from 0.25°×0.25° daily gridded rainfall datasets as in the National Data
Center, India Meteorological Department (IMD) (Guhathakurta and Rajeevan 2008; Rajeevan
et al., 2006, 2008) during the period of 1901-2015. Owing to its better temporal and spatial
resolution, the IMD rainfall dataset has been used in several research attempts in the recent
past for analysing the morphology of drought occurrences over India (e.g. Sinha Ray and
Shewale, 2001; Gore and Sinha Ray, 2002). In previous literatures there have been various



mentions for identifying certain days as dry, based on some predefined daily rainfall
accumulation thresholds. Singh et al. (2009) has mentioned that days having rainfall less than
5mm/day can be considered as dry. But this criterion is only valid for ecological droughts and
hence it will not be a suitable threshold for many Indian regions experiencing very low
rainfall. Recently, another classification scheme has also been attempted by Said et al.,
(2014) where rainfall accumulation lower than 1 or 3 mm/day is considered as a dry day. So,
to further check which threshold provides best results, the correlation coefficient of DI verses
DDF are plotted in **Table 2**. The correlation coefficients follow some spatial diversity but
interestingly, they do not exhibit much change with respect to the rainfall threshold. Hence,
to understand its implication, the number of days having rainfall accumulation above 1 and 3
mm (during JJAS) is expressed in the form of ratio in **Figure 1**. The ratio indicates that for
all the months and regions, days having rainfall accumulation above 1 mm/day are more in
number compared to the days having rainfall accumulation above 3 mm/day. This makes it
reasonable to put 1 mm/day as threshold rainfall accumulation for DDF consideration as it
will filter out only the intensely dry conditions which will make the drought identification
more reliable. Hence, this study is progressed using 1mm/day as the dry day identification
threshold. Further, the DI values obtained are normalized with respect to mean and standard
deviation for simplicity. Data sets of number of dry days and drought index are passed into
three dependence tests: first, using three equal sized grouped box whisker distributions;
second by principle component analysis of variances of two main contributors. The third and
final approach involves a multi-linear regression in order to see the net contribution of the
various components on dry or wet condition,
Datasets of sunspot numbers are considered here as a reliable representative of solar
activity, which in turn may modulate the earth's hydrological balance; hence utilized in the
current study. Monthly averaged sun spot numbers are obtained from the Solar Influences
Data analysis Center (SIDC) in the Royal Observatory of Belgium from the year 1749 till
present (Cliver et al., 2013). This study also considered ENSO index, obtained from the
Oceanic Niño Index (ONI), which is calculated using 3 month running mean of Extended
Reconstructed Sea Surface Temperature, Version 5 (ERSST.v5) SST anomalies in Niño 3.4
region ($5^{\circ}$N – $5^{\circ}$S, $120^{\circ}$ – $170^{\circ}$W) with a 30-year base period (Huang et al., 2017). Conditions
resulting in values beyond the threshold of ±0.5°C are considered to be either an El Niño or
La Niña. These datasets are obtained from 1950 to present. Present study also uses
0.5°×0.625° gridded datasets of  AOT at 550 nm, Black Carbon (BC),  dust (pm2.5 only),



Organic Carbon (OC), sea salt and sulphate obtained from MERRA-2 (Modern-Era
Retrospective analysis for Research and Applications version 2) provided by NASA.
MERRA-2 provides global reanalysis product since 1980 to present
(https://gmao.gsfc.nasa.gov/reanalysis/MERRA-2/). Reliability of all the aerosol products
from MERRA-2 can be found in Buchard et al. (2017) and Randles et al. (2017). Out of all
the aerosol components mentioned, only black carbon concentration datasets are found
available for validation against Aethelometer measurements over Kolkata. However, to
preserve the parity with monthly averaged Black Carbon Extinction as in MERRA2, the
observation datasets are also monthly averaged for a net period of 36 months during 2013,
2015 and 2017. Consequently, a well matching is observed between the two sources as shown
in **Figure S1**. To be double sure, the datasets of BC AOT and concentrations are both
normalized and then their probability distributions are plotted. The distributions fitted with
Gaussian curves shows almost similar behaviour in both the cases, which shows the
suitability of this datasets in subsequent sections.
In addition, the ERA Interim reanalysis cloud cover data is utilized
(http://www.ecmwf.int/) at 0.75°×0.75° default resolution (Beriford et al., 2011). As DDF is
being observed mostly over the month of August, hence monthly averaged data of total, high,
medium and low cloud covers are extracted over the required regions and are plotted for the
same time period (as in for aerosol parameters) during 1980-2015. The idea behind using this
dataset was to identify the association between increased cloudiness and reduced rain
accumulation during the mid-monsoon months. Additionally, in order to show its relation
with cloud microphysics, dataset of cloud particle radius are utilized, and is obtained from
NASA Earth Observation (NEO) portal
(https://neo.sci.gsfc.nasa.gov/view.php?datasetId=MODAL2_M_CLD_RD). The dataset
provided by Terra/ Aqua Satellite of MODIS on daily, weekly or monthly basis with a good
spatial resolution of 1°×1°, and is available only over a relatively shorter span of 2000 –
2018. The monthly averaged values of CER have been utilized during the month of July and
August for the present study.
This study uses gridded population density (as a proxy of urbanization), obtained from
Gridded Population of the World (GPWv4), and provided by the CIESIN-SEDAC database
from Columbia University for the year 2000, 2005, 2010 and 2015. This data set is
constructed by extrapolating the population data from national or sub-national administrative
units all around the world. The resolution of the product is 30 arc-seconds, or approximately





1 km at the equator, further details about the data can be obtained from
http://sedac.ciesin.columbia.edu/data/set/gpw-v4-population-count-rev10.

**3. Results and Discussion**
**3.1. Identification of potentially drought prone regions over India**
Considerable conditions for drought occurrences are identified on the basis of the
balance between monthly PET and rainfall accumulation during June-September as depicted
in **Figure 2**. It is seen that due to arid climates, north western India experiences higher values
of PET particularly up to July which may happen due to late arrival of monsoons at that
location and hence this region may be considered for the analysis. On the other hand the
south eastern peninsula of India experiences higher PET values, hence it has been considered
for further analysis. However, the rest of the country experiences much lesser values of PET.
In contrast, precipitation values are consistently lesser both in the north western India as well
as the south eastern peninsula, so both these regions may face more probability to experience
negative DI, hence are selected for analysis. Another highlight from the figure is that, the
mid-section of IGP depicts a sharp gradient of precipitation. This diversity becomes more
prominent during the months of July-August as during this period, the entire IGP experiences
very heavy rain accumulation (>300 mm on average) but the mid-IGP experiences much
lesser rainfall ~200 mm. Consequently, this mid IGP region is also selected for analysis.
Accordingly, the grid points with 0.5 degree resolution in these three regions are identified
and accumulated to form three main study regions which are numbered 1, 2 and 3
corresponding to IGP, South Eastern peninsula and North West India, respectively as shown
in **Figure S2**.
**3.2. Importance of dry day frequency (DDF) in analysing the drought conditions**
After the identification of the drought prone regions, the main objective is to
determine a suitable parameter which best represents the probability of droughts and which
also can be related to other natural and anthropogenic factors in all regions. Hence an
assumption is taken, that if the temporal distribution of rainfall is considered constant month
wide, then a drought is only possible when both PET is high and precipitation is low. Now
low precipitation and high PET mainly arises from multiple dry day occurrences in a month
leading to droughts. So, for simplicity, during each of four months in three seasons, the
difference between precipitation and PET is calculated over 115 years and the obtained data

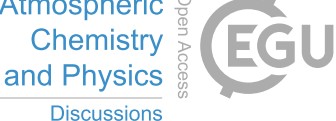

is normalized with respect to mean and 1 sigma std for simplicity, Next, the DDF time series
is calculated from daily precipitation values as already described previously after which the
correlation coefficients between drought index DI and this dry days frequency are calculated
and shown in **Figure 3**. The correlation analysis is done for two overlapping periods of 115
and 60 years namely: 1901-2015 and 1956-2015. The reason for this two part analysis is that
during the second part, more technological advancement may lead to more reliable daily
rainfall data, this is because during recent years the advent of more accurate rain gauges have
led more reliability in deciding whether daily accumulation < 1mm and thus more reliable
dry day frequencies are calculated. Another reason is that, second part witnessed more station
and satellite data sources, so possibility of relationship is expected to be stronger in last 60
years. However, to bear better evidence to the above stated hypothesis **Figure S3** shows
scatter plots of DI and dry day frequencies for all regions and months.

The correlation values for region 1 and 3 during both 115 and last 60 year span are

depicted in Figure 3. Reasonable correlation coefficients are obtained in both regions 1 and 3
over 115 years. Importantly, better correlation values are observed typically over July in
region 3 and August in region 1while it is lesser in all other cases. This is because, regions
situated in the western and north western parts of the country (mainly region 3) generally
experiences delayed monsoon as supported from many independent sources which may lead
to correlations in June and July in region 1 and 3, region 1 specially shows good correlation
in August which is mid monsoon month which need more attention in coming sections.

Considering the last 60 years, correlation coefficients improve in all regions and

months as expected. Region 3 shows high correlations in July followed by August, while
region 1 depicts comparatively much higher values during July and August. Thus the
consideration of delayed monsoon onset may bring out more dry days in region 1 and 3. But
on the other hand region 1 shows a high association between DI and dry days in August
which needs to be studied. Region 2 is mainly influenced by precipitation occurring during
the late monsoon months i.e. September and not by the mature monsoon stage which is
evident from the higher correlation values at that time. Hence this region may not fit with the
scope of the present study. Additionally, as more significant correlation values are obtained
during 60 years span compared to 115 year scale, hence DDF trends will be studied over  the
last 60 years span in the coming sections.



### 3.3 Determining the time spans for the analysis of DDF trend over mentioned regions

### 3.3.1. The importance of partitioning Region 1 for further analysis

It can be seen from the preceding sections that the correlation between DI and dry days for region 1 is noticeable but it is not highly prominent due to the presence of many outliers in the scatter plots (Figure S3). This is because region 1 encompasses a total spatial coverage of $5^o X 8^o$ which has a lot of topographical and climatic diversities between them. A better example of this has already been depicted from the precipitation diversity in Figure 2 where the precipitation gradient was found to change abruptly even within region 1. So these spatial diversities can interrupt the association between droughts and dry days. Hence to have more realistic investigation, the region is now partitioned horizontally along $81.25^o E$ which lies in the middle of IGP. This gives rise to two different regions in the east and west of region 1 which will hereafter be referred as region 1a and 1b, respectively. Next the total distribution of data of DDF for 1a, 1b, 2 and 3 are again investigated for two overlapping periods 1901-1960 and 1956-2015 for parity in **Figure 4(a)**. Region 2 and 3 show almost no change in the distribution before or after 1955, hence it is not given importance. Region 1b shows sight increase in mean and median but with no prominent change in the distribution while the same thing is very prominent over region 1a. In this case, since the last 60 years the mean and median values changed by more than 4 days which is a very alarming fact. Most importantly, the upper quartiles and whisker have ascended to a maximum value of 30 days, which indicates severe drought occurrences. Thus it can be inferred that region 1a emerges as a prominent drought prone region showing an abrupt rise in DDF especially over the month of August and hence it will be investigated in detail in the coming sections.

### 3.3.2. Analysing the climatic trends of DDF using a 15 day window

It has already been discussed that the drought intensity has significant correlation with DDF On a monthly basis. However, it is also necessary to investigate whether the intra-monthly distribution of rainfall may also have its own impact in modulating the dry day frequencies especially during the mid-monsoon months which experience maximum precipitation variability. Hence, the monsoon months (JJAS) are now divided into 8 equal slots of 15 days each and the 60 year time series for all these regions are obtained. Next the robust-fit trend analysis at 95% confidence level is done to find the mean yearly trends, which is multiplied by 60 years and then normalized with respect to mean to generate a percentage wise change in DDF.





The percentage changes are shown in **Figure 4(b)** which depicts an overall increase
of DDF for all regions with a few exceptions. Region 2 shows very weak trends (< 5%) all
throughout monsoon, however, by the end of September, a reasonable trend of ~20% is seen
which may link to dry phase developments in the later months.  However, this period falls at
the declining phase of monsoon which is beyond the main scope of this study; hence it may
be neglected.
Region 3 shows quite weak but alternating dry day trends over June followed by the
month-long increase in July. This indicates a probable change in the timing of monsoon
rainfall over region 3. However, this cannot be firmly confirmed as there is no particular time
slot having a prominent trend value (all cases showing trends < 5%). Rainfall in June and
early August lead to dry region conditions over July, but the cumulative monthly growth in
July is ~10% which is not very strong enough and hence it will be discussed later in the
study.
Finally, Region 1 shows very strong increasing trends in dry day frequencies with
similar pattern over 1a and 1b. Both these sub-regions experience relative wetting at late
June, followed by a prolonged dry phase up to September. But the main difference between
the two sub-regions is that is that the trends are consistently high all throughout in 1a with as
much as 60% and 20% increase over August which also continues onto September; while in
region 1b the trend values are comparatively lesser (40% and 5%) during August. Thus, it can
be inferred that though a clear increase in DDF is obtained all throughout region 1 during
July-September, yet the trends are relatively stronger in region 1a especially during August
which demand primary importance throughout the study.
**3.4 Analysis on DDF trends over region 1a**
**3.4.1. Investigating the probable influence of natural and anthropogenic components on**
**DDF for region 1a**
In light of the previous sections, the probable influences behind the increasing trends
in dry day occurrences are investigated over region 1. Number of natural or anthropogenic
factors may be responsible for this phenomenon. While natural factors mainly include the
effect of solar activity, ENSO variability or moisture tendencies, the anthropogenic
constituents mainly include aerosols which again encompass a lot of organic and inorganic
pollutants. Now, to quantify the effect of aerosols, the aerosol extinction coefficient values
can be utilized from either satellite observations (MISR) or from dedicated model simulations



(MERRA2). Since observational datasets from MISR satellites are very sparse during
monsoon season and also the total measurement period is only 16 years, hence MERRA 2
datasets are used for further analysis. Keeping the availability of AOD datasets in mind
further analysis has to be concentrated on 36 years span between1980-2015. Owing to the
prominence of DDF trends during the month of August, further studies are concentrated on
this period only. As already mentioned natural factors like solar activity and ENSO
oscillations (hereafter referred as SSN and ENSO) may have some impact on precipitation
variability which is also supported from previous attempts taken, hence they are considered.
Additionally, moisture content also directly controls precipitation and so their monthly means
at 850 hpa (corresponding to maximum moisture content during monsoon) are also utilized
from MERRA 2 reanalysis database. To understand the dependence of these factors on DDF,
first, the monthly DDF values during August 1980-2015 are arranged in descending order and
then the sorted dataset is divided into three equal groups as Short Dry Phase (SDP)
corresponding to normal conditions (8-10 days with average of 9), then Medium Dry Phase
(MDP) signifying near drought (10-14 days, with average of 12.5 days) and Long Drought
Period (LDP) which represents a full drought conditions (14-18 days, average ~ 16) as
depicted in **Table 3** and they are also hereafter mentioned as SDP, MDP and LDP. Next, for
all these three groups, the distribution of total aerosol extinction (AOT), SSN, ENSO index
and SHUM at 850 hpa are shown in the form of box plots in **Figure 5(a)**. It is seen that as
DDF increases, the distribution of total aerosols start increasing, as evident from the rise in
median and upper whisker values. The variation of SSN is almost random in all cases hence
neglected. Additionally, ENSO intensity changes fairly with droughts. The upper whiskers
and median rises slightly, but its effect is doused due to a dominant overlapping between the
groups which fails to indicate a clear relationship. Specific humidity shows a minor decrease
in all groups, though the median and quartiles do not show any prominent change (from 15 to
13 g/kg). Hence the importance of this factor cannot be ascertained.

As the dry phase length distribution fails to identify the dominant factor behind the

rise of dry days in region 1a, hence all these four factors are passed through principal
component analysis test (PCA) and the results are shown in **Figure S4(a)**. The analysis
produced a set of three orthogonal components out of which pc1 and pc2 account for 50 and
25% of variances so we can neglect the contribution of the $3^{rd}$ component. Next, the
corresponding variance scores of these components are plotted in Figure 5 which shows that
SSN and humidity have very less variance according to the pc1 axis hence considered as less



important, but aerosols and ENSO have comparatively higher values so they should be considered important for further analysis.

Further, multi linear regression analysis is done to see the independent contribution of these four parameters to DDF. All datasets are normalized so as to get uniform variability to enable easy identification of the dominating factors. The MLR concludes that the coefficients for aerosol, SSN, ENSO and SHUM are 0.393, 0.008, 0.161 and -0.207 as shown in **Table 4**, SSN does not show any effect hence finally rejected. ENSO and specific humidity have significant contributions but in opposite manner and also their distribution analysis showed significant overlapping; hence they should not be considered in order to remove ambiguity. Finally aerosols have a coefficient of 0.393 which is much higher than the others as also observed in the PCA test and distribution analysis. Hence one has to consider aerosols as the more dominating factor compared to the other natural components in modulating the dry day occurrences.

**3.4.2. Significance of various aerosol components influencing the DDF over region 1a**

Total columnar extinction values of 5 aerosol components namely: black carbon (BC), Dust PM2.5, organic carbon (OC), Sea Salt and Sulphates are obtained from MERRA 2. BC and OC mostly comes from anthropogenic sources and significantly contribute in warming up the atmosphere. It has been reported in earlier studies that the presence of BC aerosol in rain cloud may have "burn off" effect on the cloud due to heating [Ackerman et al, 2000, Babu and Moorthy, 2002]. On the other hand aerosols like PM2.5which may have both natural and anthropogenic sources can also influence the cloud life time by increasing cloud droplet number (Zhao et al, 2017; Sato et al, 2018). Thus, the cloud coverage is modulated and precipitation process is affected. Now the change in concentration of these parameters during last 36 years over region 1a has been discussed in the next section.

Though it has been discussed in the previous sections that aerosols have a dominating influence over dry day occurrences, however, it is yet to be specified which type of aerosols (natural or anthropogenic, organic or inorganic) are becoming major influencing factor for this phenomenon over region 1a. Hence time series datasets of these five components are again taken for 36 years and are grouped with respect to the corresponding dry day ranges as already explained in previous section. After that the corresponding distributions are plotted in box plots in **Figure 5(b)**. The distribution analysis depicts that the sea salts show some overlapping which reduces the impact on DDF. Sulphates have quite high values all





throughout but their medians or distribution does not exhibit any deterministic sequence (first
decrease then increase); so they also cannot be used here. Dust AOT values are less but its
median shows weak contribution towards drying, but the overlapping in distribution makes
the overlap association very weak. But compared to the others BC and OC have shown a
better association with DDF along with reasonably increasing tendencies in medians and
quartiles. But this phenomenon also hints towards a dominant component of pollution coming
from certain highly urbanized sectors of region 1a such as Lucknow, Allahabad (25.43° N,
81.84° E) and Varanasi(25.31° N, 82.97° E). Again out of these two, BC has relatively better
variation as it has the least overlapping nature so it may be considered the most dominant
factor. But still to have better evidence, the PCA and regression analysis are attempted.
The PCA analysis results are depicted in **Figure S4(b)** which shows the contribution
of pc1 alone is 60% followed by pc2 of 25% to be more prominent hence there may not be a
need to study pc3 here. From the scores it is found that sulphate and dust behave similarly in
their variances with high pc1 and low pc 2 values, but OC and BC have both high pc1 and
pc2 components, so they may be found responsible for the variability in dry day changes.
However, sea salt also may have some influence but it is not much clearly understood from
the figure.
To clarify any remaining misconceptions, the MLR coefficients are computed which
gives the values as 0.542, 0.129, 0.263, 0.326 and 0.124 (shown in **Table 5**). It is expectedly
obtained that the dust and sulphate have very less contributions so should be neglected. BC,
OC and sea salt have higher values, of which OC and sea salt have comparable magnitudes,
but, sea salt has much less AOT values with lesser pc1 variance score and also reasonable
distribution overlapping, so the effect of OC  may be considered better. BC has very high
MLR coefficient with high pc1 score and also a clear variability of distributions. Hence, it
may be concluded that owing to urbanization, the effect of BC followed by OC has much
stronger association with drought intensity and dry day occurrence.
**3.5. Analysis of DDF trends over Lucknow**
From the previous section, it has surfaced that urbanization may have a dominant
association with the increase in DDF during August. To be definite about this, a re-
investigation has been done over Lucknow (26.8°N, 80.9°E) which is the state capital of the
state Uttar Pradesh, and is a more urbanized point location belongs to region 1a. However the
relationship of DDF with SSN, ENSO and SHUM is not shown as Lucknow already falls in
region 1a whose synoptic effect would not change within the region. Only, here, the effect of
individual aerosol components is also depicted in the distribution analysis as shown in **Figure**
**6**. Now, in case of Lucknow the variability in dry day values are much stronger as shown by
SDP (4-12 dry days average 9.5) MDP (13-17 days with average 15) and LDP (18-30 days
with average at 22 days) mentioned in Table 1. The distribution analysis on total aerosol
AOT shows much larger values over Lucknow than in region 1a and also the variability of
the median values with the quartiles and whiskers are also far more deterministic here which
may have influenced the entire distribution towards more dry conditions. Next, coming to sea
salts and sulphates, they have much less values than in region 1a due to its significant
distance from the seas. Sulphates show no meaningful variation, hence are rejected
straightaway, sea salt values are less but the variation of median and upper whisker shows a
prominent increase which may be important. However, the lower quartile is very small and
overlapping in all three cases which serve as a setback to its variability. However, Dust does
not such variations due a considerable overlapping in it. But on the other hand, BC and OC
do not have much overlapping and they also have clear increase in medians and both quartiles
thus supporting the more sensitivity of this region towards dry days.
**Figure S5** shows the distribution analysis of these components with PCA tests. The
analysis reveals the presence of three strong principal components where pc1 is 60% and pc2
of 30%; hence pc3 is not considered further. Next, when the variance scores for these
parameters are plotted, then all factors show almost similar values of pc1 score, so pc2
becomes important. While judging the pc2 scores, we see that BC followed by OC has the
best variability in this set hence they may be considered for the dry day variation. To confirm
this, multi linear regression is done on the components and the results yield values of 0.864,
0.218, 0.556, 0.0106 and 0.155 (Table 3). According to previous results, the contribution of
BC and OC is much higher than the others, with BC showing a higher correlation in all cases
compared to OC, hence the dependence of dry days can be primarily associated with
urbanization. Dust follows this parameters but its dependence is comparatively much smaller
than both BC and OC which further supports these findings.
**3.6. Comparative analysis on the DDF trend of last 60 years and Cloud properties**
**among Region 1, 1a and Lucknow**
The preceding sections, have given an idea of how urbanization is influencing the
evolution of dry day occurrences. But to understand quantitatively its climatic impact now the
averaged DDF of last 60 years are plotted for regions 1, 1a, Lucknow. In order to examine the



change in DDF patterns as one downscales from a broad synoptic scale (IGP) to a small
localised urban location. **Figure 7** reveals that region 1 has a weak but discernible increase
from 5 to 15 days in last 60 years. When robust-fit analysis was performed, it was inferred
that the net change in dry day frequencies over region 1 is ~35% with respect to the 60 year
average. However, the existence of some periodicities in the data was observed while no
evident extremes were observed in the time frame. The value of the slope is found to be less
(0.074) which leads to a poor r of 0.384.  For region 1a the total variability is from 5 to 18
days; so the slope is expected to improve a bit (with a robust-fit net trend of ~44% with
respect to the average) while the periodicity seems to be apparently disturbed due to presence
of more data extremes. Finally, in case of Lucknow, huge change is observed from 4 to 25
days which indicates a complete shift in rain climatology with trend values as high as 61%
with respect to 60 year average during August when normally, the maximum rainfall occurs
over India. Huge number of outliers and extremes are seen some of which are close to 30
days (indicating no rain over August at all). The periodicity also seems to be disturbed due to
outliers resulting in a very sharp slope of 0.139 per year. Thus the severity in drought
climatology is well explained with respect to urbanization as already hypothesized earlier.
But it may be noted that the increasing trends and correlations are mainly caused by more
occurrence of high dry days in present rather than a gradual rise in the mean values;
additionally there are also some periodicities in the signal which results in the correlation
being less than 0.5.

It is reported earlier that increase in anthropogenic aerosols may lead to more number

of CCN causing reduction in cloud particle radius (**Figure 7**) which may result into less-
occurrence of rain in spite of the increase in cloud cover. From previous section it is clear that
dry day frequency exhibits a definite increase in magnitude over region 1a and Lucknow.
Since anthropogenic components have shown highest possible dominance on dry day
occurrences, so an attempt is made to identify how cloud parameters has changed with time
over region 1, 1a and Lucknow having different urbanization growth and so on the
anthropogenic components. Region 1 which is covering a broad area does not show
prominent change in DDF and it is also observed that that the change in cloud cover over
region 1 (~ 2%) and reduction in cloud particle size are very feeble. But interestingly as the
region of concern is downscaled to Region 1a followed by a further downscaling to a region
the urbanization impact becomes prominent and that is also reflected in the observed cloud
parameters. It is evident from the figure that there is a decrease in cloud particle size by 6.4%



over region 1a only in last 19 years. This has significantly increased the cloud lifetime
resulting in a more definite growth of mid and low level clouds. The situation however,
becomes more prominently worse in case of Lucknow where cloud particle radius shows a
decreasing (12%) trend in last two decades and accordingly cloud cover increased
consistently (~18%) reflecting the impact of urbanisation. As a consequence, the dry day
frequency ascends at a rapid rate over Lucknow in spite of increasing cloud cover which
definitely needs to be studied in more detail in future approaches.

The long term trends of dry day occurrences have exhibited a prominent growth in dry

days but the effect of this trends were found to be subdued to some extent by several
periodicities over the last 60 years in both region 1 and 1a. To understand their role to a
quantitative scale, periodicity analysis is done on last 60 years using autocorrelation functions
and the results are depicted in **Figure S6**. The ACF values show highest value of 1 for a time
lag 0, hence it is removed. Also there is no use in understanding periodicities greater than half
of the period hence the maximum period is fixed to 30 years. 1 sigma bars are provided to
understand which periodicity may be significant enough to impact the long term trends. The
figure shows that the ACFs are reducing with time for all regions just as expected. However,
only two points are found considerable, one is at the local maxima of 4 years corresponding
to ENSO, where as expected the synoptic influences will be stronger in larger spatial scales.
Another periodicity is expected to lie at ~1-2 years which represents the year-year varying
component of urbanization. However, this effect is found to be much lesser in region 1 as it
has a much higher spatial scale. But in case of region 1a the 1 year periodicity is expected to
more prominent than in region 1 which is also supported with the comparatively lesser
contribution of ENSO in region 1a as also shown. Again, because of the same reason, the
year to year variability (shown by periodicity 1) should be most dominant in Lucknow
followed by 1a and then 1. The same thing follows in the figure and interestingly, the effect
of urbanization overshadows the other factors like ENSO in the periodicity analysis for
Lucknow (due to presence of many outliers) as shown previously. The contribution of both
outliers extremes with periodicities are seen almost comparable in region 1a. But in region 1
the effect of periodicities is more than the outliers as clearly seen with higher ACF in ENSO
for region 1 compared to 1 year periodicity case. This clearly infers about the effect of
urbanization which suppresses the effect of ENSO periodicity and thereby results in the
drastic increase in DDF over Lucknow.



### 3.7. Analysis of DDF trends over Region 3

### 3.7.1. Probable influence of natural and anthropogenic components on DDF for region 3

In most of the preceding sections, the variability of DDF has been studied over Region 1 falling in the IGP. However, the north-western part of the country also comes under high drought severity zone as already discussed; hence this region is studied in detail now. Figure 4 has showed that the DDF trend is comparatively higher during the month of July; hence DDF during that month will be considered hereafter for further analysis over region 3. But it may be noted that the change is not so much prominent here as in region 1 (with a cumulative average of ~8% rise) and also the yearly fluctuations are too large which has subdued the trends resulting a feeble rise of two days in the last 60 years (23-25 days) over this region shown in **Figure 8**. To start with the distribution analysis, three classes are made as SDP (14-20 dry days average 19) MDP (21-24 dry days average 22.6) and LDP (24.5-27.5 days with average 26 days) as depicted in Table 2. It may be noted that the values themselves have high magnitudes for all classes and the variability is also quite less (19-26 days) here compared to 9-22 in Lucknow; so the observed variation also should not be much prominent which is also evident from **Figure 8(a)**. Further, as this region generally experiences arid climate, hence specific humidity can be an important factor here. Accordingly a decreasing trend is seen as supported by the median and lower bounds. But there is more overlapping among the classes and the total variance of humidity at 850 hpa is only between 12-10 which may not be strong enough to modulate drought intensities all by itself. SSN shows no definite variation hence not considered further. Aerosols and ENSO seem to have a weak increasing trend in their medians which again is diffused by more overlapping in these distributions. So this weaker variability is in good agreement with the feeble trend in dry days, but simultaneously makes it difficult to determine the potential driving factor behind the increasing DDF in region 3.

A better insight into the inter-dependence of all these components are investigated by the PCA test in **Figure S7 (a).** The analysis reveals four PCA components out of which three PCs are considered to explain the complete range of variances in dry days. The scores signify no definite pattern with the total aerosol AOT assuming high pc1 and low pc2 pc3 while ENSO has high pc2 and pc3 with lesser pc1 and SHUM falls in completely different quadrant. Now since aerosols have higher pc1 component which is comparatively stronger than other pcs so it may be a deciding factor. To clarify this confusion, MLR coefficients are calculated which come around 0.107, 0.078, 0.056 and -0.267 also shown in Table 2. It is





clear from the MLR outputs that specific humidity has a strong negative influence on dry
days so it will have good effect on drought occurrences. But apart from this, the second
dominant factor behind droughts is still found to be the aerosols. However, this fact needs to
be supported with more detailed analysis as shown in the later sections.
**3.7.2. Analysing the influence of different aerosol components on DDF for region 3**
The distribution analysis of aerosol components are now shown in **Figure 8(b)**
which depicts that as usual, sea salt aerosols and sulphates have no role in modulating the
DDF. But it may be noted that here the magnitude of sea salts and sulphates are higher than
in region 1 or 1a may be due to its transport from the nearby seas which has not been washed
away by rain in its path owing to the arid climate. However, experience a very prominent
overlapping between the components which reduces the overall trend. The variation of OC is
not clear and hence is obliterated. BC as usual has a deterministic variance with some
overlapping; but still the whiskers and median values indicate its impact on dry days. Another
important aspect here is that, the range of values for these parameters are much lesser here
due to lesser urbanization which still affects the DDF. But the contribution of dust aerosols
emerges as the dominant component here as it not only shows higher values compared to all
other regions but it also signifies a clear trend in the medians and distribution values. Thus it
can be inferred that both dust and BC may contributed to this phenomena.
To investigate which parameter has more dominance in dry days formation, PCA
analysis is done on the individual components and the results are depicted in **Figure S7(b)**.
Here four PCAs are obtained, but the first two PCAs contribute 80% of variability so the 2D
variance is seen. Also the contribution of pc1 is comparable to pc2 so here both will be
important. While analysing the scores it is observed that only dust and BC have both high pc1
and pc2 so should be considered while most of the others have lower pc2 scores so they can
be neglected. Further investigation is done on MLR analysis towards the trend contribution
which also gives similar outputs as 0.464, 0.431, 0.120, 0.182, and 0.033 (Table 3). Again
here both BC and dust emerge potentially significant for the region 3 to be considered in
associated with the slant rise in dry days. Both of these two components may have local
sources but owing to its location, there are possibilities of having added amount of dust
aerosols being transported from adjoining deserts or from dust storms and fumigation of dust
from the ground during intense dryness which are not found prominent over the region 1a
(where BC and OC was high due to high urbanization). Further for more meticulous



observation we have also examined the cloud particle radius and cloud cover (**Figure 9**)
which shows that all four types of cloud cover have remained almost unchanged over the
years and there is a weak reduction in CPS (~2%) unlikely to the region 1a (6.4%) or
Lucknow (~12%). This is again in good agreement with less prominent increase of
anthropogenic emissions or in short less increase in urbanization over region 3 compared to
region 1a or Lucknow. This is further discussed in coming sections. But few things are
important to mention here: the trend of dry days in region 3 though it is weaker compared to
region 1a may have serious impact in future as the region already experiences high number of
dry days itself so a slight increasing trend is also alarming. Thus the effect of urbanization
will be still an important parameter contributing towards the hike of BC and (some of) dust
aerosols growth and in turn leading to more strong trends in DDF over this region.
**3.8. Impact of urbanization on DDF trends**
From the previous section, a strong association has been obtained between dry day
occurrences and urbanization due to high BC and OC or dust. Now to prove whether it is due
to urbanization, one needs to study the effect of land use or vegetation cover. But these
datasets are either not available in public domain or their reliability is not good enough. On
the other hand high population density at a location is generally associated with the growth of
urbanization. So, this concept is utilized from gridded $1^{o}$ population densities during 2000 -
2015 from the SEDAC website. The primary distribution of population for year 2000 is
shown in **Figure S8** which depicts, more values at region 1a compared to region 1b, and
another thing is that, Lucknow is still found as a patch of very high population even at 2000.
On the other hand, region 3 had much lesser populations at the same time. Next, the trends of
population density are observed over region 1, 1a and Lucknow and the results are depicted
in **Figure 10**. It is observed that all throughout region 1, population density rises from 650-
800 persons per sq kilometre which quite a high value is. Next, region 1a shows much higher
values than 1 with a steep rise of 760-1000 persons per square km. So region 1 has
consistently high population average and trend will definitely lead to higher OC and BC. The
situation worsens in Lucknow where population changes from 850-1100 persons with most of
change happening in last 10 years which strongly supports the amplified effect at Lucknow
compared to 1a. But region 3 shows a very less value comparatively from 100-140 leading to
less BC OC there, but relative change there is 40% compared to Lucknow (30%) so in future,
if urbanization and population persists to grow there in this rate then this constituents of BC



and OC with dust will grow to alarming limits which can cause drastic change in DDF over
North-Western Indian regions.

### 3.9. Probable contribution of air mass transport over region 3

From the previous section, it follows that urbanization has considerable impact in
increasing the dry phases over region 1a during the mature monsoon phase. But in region 3,
relatively the effect of urbanization is feeble as has been reflected through the less population
density and BC, OC concentrations. However, the observed increasing trend of dry days in
association with the increase in dust aerosols over this region may be partially attributed by
the loading of dusts aerosols from local sources and partially transport from the adjoining
deserts. To investigate the transport issues the back trajectory analysis has been carried out
during second week of July (shows highest trend in dry days over region 3) using HYSPLIT.
The frequency of all possible trajectories are drawn in 12 hour steps for the preceding 10 days
at 1 degree resolution of GDAS data at 2000 m to understand the probable transport of dust
aerosols. The endpoint of the trajectories has been taken fixed at $27.5^o$N $72^o$E (pointing to the
centre of region 3). Primarily, trajectories are drawn for all available years from 2005 – 2018
but observation does not lead to any significant inferences because the trajectories show a
wide range of variability. Next, to understand them in more precise manner, an attempt has
been done by considering three sets of years having too high number of dry days (2009, 2015
having dry days > 23), moderate number of dry days (2010, 2014 DDF~20) and less number
of dry days (2011-2012 with DDF<16) with similar population and meteorological pattern.
After that, the frequency of back trajectories for each of these 6 years is depicted in **Figure**
**S9**. No noteworthy similarity is found observing the trajectories for different set of years
indicating any prevailing paths of air mass transport. Though it may be noted that the arid
land mass of Afghanistan and Middle East may have some contribution in transported dust
aerosols (as the figures show mostly significant air masses path from west and from North
West) but it is not enough to confirm any dominant path of air-mass transport in region 3
indicating any clue of increased loading of dust aerosols.
Further, for more confirmation all the back trajectories available during the June-July
are accumulated for 2005-2018. For each of these years the frequency distribution of
trajectories is accumulated with respect to hour lag from -1 to -120 hours and grouped into
five classes in such a way that the first group consists of all possible trajectory endpoints
throughout the last 24 hours before arrival accordingly the second group represented 24 to 48
hours before arrival and so on for the five days. The latitude and longitude corresponding to



the endpoints for each day trajectories are recorded and parsed into five groups to plot their
frequency distributions shown in **Figure 11.** It can be noted that in day 1 the latitude or
longitude distribution is confined within a very thin spread which gradually diverges with
days. From day 3 the spread maintains a band of longitude span around 60-75 E and Latitude
spans from 20-30 N covering most of Pakistan and a portion of Arabian Sea. Further in day 4
and 5 the spread of distribution becomes more diverged (Lon-55-75, lat 18-35) covering the
Middle Eastern Asia to the north Arabian Sea. Overall, the source points of the transport of
air mass are too random and insignificant to attribute in the increase in dry day occurrences.
This again suggests that also in region 3 local sources and urbanization influenced DDF but
only with a lesser impact compared to region 1a.
**3.10. Future trends of DDF over Region 1 and 3 using RCP 8.5 scenario**

The next concern of this study is to investigate the projected change of dry phase

lengths over the foreseeable future. Many attempts in the recent years have employed CMIP5
GCM simulations to provide future projections for any urbanization scenario. In accordance
with the present study, RCP 8.5 projections of rainfall (and DDF) corresponding to maximum
urbanization levels has been considered over the mentioned regions. It may be noted that in
the last sixty years itself, DDF values have reached ~ 30 days in August, hence it is useful to
study DDF in a two months span of mid-July to mid-September (having a reasonable
increasing trend in dry days). The future projections of DDF over this time span is now
obtained from 1950-2100. But the reliability and accuracy of these datasets first need to be
validated from in-situ measurements. Hence, historical daily precipitation datasets of r1i1p1
realization from 11 well known GCM simulations are taken during 1955-2005 for all grid
points in region 1 and 3 after which the DDF is calculated and recorded. Finally the averaged
DDFs from each model was compared with the IMD data and the correlation coefficient with
the normalized standard deviation values in **Table 6** indicate that three models namely: CAN
ESM2, CNRM CM5, NORESM 1M show better agreement; hence they can be utilized to
generate future projections for region 1 and 3 up to year 2100. For simplicity the yearly
means of DDF historical data from the models are also shown in **Figure S10** which again
are found to follow the expected trends of DDF in all three regions

Next the total variation in dry days are investigated over region 1 and 3 including both

historical and CMIP5 RCP 8.5 projections data to get a 150 year trend of dry day frequencies
in **Figure 12**. The DDF for all 29 grid points in region 1 and 20 grid points over region 3 are



averaged yearly and then depicted in Figure 7 and 9. The multi model mean data shows that
even when averaged spatially, dry days show clear increase from ~ 8 days in 1950 to ~40
days near 2100. Thus Region 1 will experience a rise in DDF from 10% to 70% during mid
monsoon phase which is highly alarming and is attributed to the rapid pace of urbanization
over those regions in the future. Again, this trend looks less discretely increasing compared to
the historical trends over Lucknow. Again, in certain cases the projected DDF is expected to
increase up to ~50 days (80%) during the 2100 monsoon which should lead to severe drought
conditions. Again, the trends look comparatively weaker in first fifty years (8-12), then it gets
stronger (12-24) and finally shoots up to very high values (24-42 days) after 2050 which is
primarily caused due to high urbanization rate over this region in the future. However, when
the same analysis was done for region 3 DDF was found to increase steadily from 20 to 40
days over 150 years. The trends of DDF are clearly much weaker in region 3 compared to
region 1 while the standard error bars are also less here. Both of this factors can be attributed
to the fact that region 3 has much less urbanization components than region 1. But it may be
noted that if region 3 continues to face urbanization at the present rate, then in future it will
experience more number of dry days. Additionally, it has been observed that, the trends have
increased almost steadily in region 3 with no abrupt change in DDF in the last 50 years like
region 1. This is attributed to the low urbanization levels at region 3 at present.
Hence region 1 creates a more alarming situation with dry days increasing by around
5 times compared to the other regions. So to further investigate this abrupt change spatially,
the model averaged data of DDF for 50 years span are shown for region 1 in the bottom panel
of Figure 12. The figure shows an expected high value around Lucknow for the 50 year
periods; but its effect diffuses as one goes towards the outskirts of Lucknow facing lesser
urbanization. Another thing is that the places adjoining Lucknow show a very drastic change
only after 2010. Thus, most of the places adjoining Lucknow shows very high number of dry
days (>45 days) near the end of this century which will grossly affect the monsoonal rainfall
leading to severe droughts and so it needs to be addressed by policy makers.
**4. Conclusions**
It is an essential aspect to study the probability of drought occurrences over India during
monsoon as agricultural and economical issues are directly related with it. Here, a detailed
study on the occurrence of dry days during monsoon over the Indian region is presented. The
study investigates three potentially drought prone regions in India based on the dearth of
precipitation and abundance of PET. Region 1 mostly belongs to the State of Uttar Pradesh




(UP), Region 2 covers major parts of the states of Andhra Pradesh and Tamil Nadu and small
portion of Karnataka while Region 3 encompass the arid part of Rajasthan. A series of
investigations are progressed which infer that over the eastern part of region 1 which is
referred as region 1a  urbanization plays significant role in increasing DDF. Prevailing impact
of anthropogenic emission like BC or OC aerosols becomes more prominent as the study
goes in depth with a downscaling approach from a broad region 1 to a specific urbanized
location like Lucknow which is one of the urbanized sectors of IGP. In association with the
increase in aerosols a reduction in cloud particle radius has been observed in our investigation
which indicates a reasonable cause of reduced rainfall occurrences and increase in DDF. This
also indicates the scope of the study over several other point locations having drought
occurrence record but could not be included in the present study approach Finally, the long
term projections of DDF are drawn over region 1a and 3 using intense urbanization scenario
of RCP 8.5 and an average of 70% rise in dry days are seen which may be a very crucial
concern by the year 2100 and hence it needs to be considered by policy makers in future
aspects. However, this study is mainly done from modelled components of aerosols, so a far
more accurate analysis can later be done over IGP subject to more availability of aerosol in-
situ data in the other major urban locations over India. The main findings of the study are
shown in a schematic presentation in **Figure 13** and are highlighted as follows:
➢ The DDF (based on the frequency of days having local precipitation accumulation
less than 1mm) has a significant level of correlation with the universally accepted
monthly SPEI Drought Index (DI) especially in the last sixty years. Further, the
correlation levels between DI and DDF are more prominent during August in Region 1a
and during July in region 3.
➢ The trends of DDF (within 15 days window) are more prominent during August for
region 1a. However, region 3 shows a descent trend during July while region 2 shows the
same during late September, (corresponding to monsoon retreating phase) hence it has
been neglected as it may not completely reflect a monsoonal drought.
➢ Results from region 1a indicate prevailing contribution of aerosols compared to
ENSO, Humidity or SSN. Further studies show that BC and OC aerosols over urbanized
region are more active in increasing the DDF, and this is also supported from
distribution, PCA and MLR analysis
➢ The trend analysis on DDF reveals that the increasing trends become stronger as the
spatial coverage is downscaled from region1 to 1a and followed by a local urbanized



location of Lucknow. About 50% increase in DDF is found in Lucknow compared to
17% all through region 1. Further, a periodicity of 4 and 8 years is found stronger in
region 1 which gets overpowered by the random urbanization component over Lucknow.
➢   Population density maps have been taken as a proxy of the urbanization component
which depict much higher values (850 persons/km$^2$ and trendsof~35%) over Lucknow
compared to the rest of region 1 and 1a. Further the population density values are very
less in region 3 (100 persons/km$^{2)}$ which is in good agreement with lesser impact of
urbanization on DDF over this region.
➢   In depth investigation revealed that urbanization components like BC or OC increase
shows significant association with the reduction tendency of cloud particle radius (~12%
reduction of CPR) and increased lifetime (~ 18% rise in LCC) over Lucknow which
results in a stronger gradient of dry day occurrences (from 9 days in 1956 to ~17 days at
present).
➢   Though in region 3 the scarcity of water vapour in its atmosphere plays a major role
to experience a high number of dry days (~23) still dust aerosols show an increasing
trend and hence it probably influences a further increase in DDF (an increase from 23
days in 1956 to 25 days at present) which is alarming for region 3.
➢   The climatic projections of dry day frequency from CMIP5 simulations of 3 GCM
model (CNRM CM5, CAN ESM and NOR ESM 1M) show a sharp increase in dry days
during July 15 to September 15 with DDF reaching up to 50 dry days over region 1 and
45 days over region 3 by 2100.

**Acknowledgments**

One of the authors (Rohit Chakraborty) thanks, Science and Engineering Research Board,
Department of Science and Technology for providing fellowship under National Post-
Doctoral Scheme (File No:PDF/2016/001939). He also acknowledges National Atmospheric
Research Laboratory, for providing necessary support and data for this work. The authors
also thank S.Jana, for his suggestions.

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





**Tables**
**Table 1 List of Abbreviations**

| Slno | Short Form | Full Form | Slno | Short Form | Full Form |
|------|-----------|-----------|------|-----------|-----------|
| 1 | PET | Potential EvapoTranspiration | 24 | CER | Cloud Effective Radius |
| 2 | DDF | Dry Day Frequency | 25 | SDP | Short Dry Phase |
| 3 | DI | Drought Index | 26 | MDP | Medium Dry Phase |
| 4 | IGP | Indo-Gangetic Plain | 27 | LDP | Long Dry Phase |
| 5 | RCP | Representative Concentration Pathway | 28 | SSN | Sun Spot Number |
| 6 | WMO | World Meteorological Organization | 29 | PCT/A | Principle Component Analysis/Test |
| 7 | SPEI | Standardized Precipitation Evapotranspiration Index | 30 | PM | Particulate Matter |
| 8 | ENSO | El Niño–Southern Oscillation | 31 | SHUM | Specific Humidity |
| 9 | IOD | Indian Ocean Dipole | 32 | CMIP | Coupled Model Intercomparison Project |
| 10 | BC | Black Carbon | 33 | AOD | Aerosol Optical Depth |
| 11 | PDSI | Palmer Drought Severity Index | 34 | MISR | Multi-angle Imaging SpectroRadiometer |
| 12 | SPI | Standardized Precipitation Index | 35 | MLR | Multi Linear Regression |
| 13 | CRU | Climatic Research Unit | 36 | BOB | Bay of Bengal |
| 14 | P | Precipitation | 37 | UP | Uttar Pradesh |
| 15 | D | Difference of P and PET | 38 | CCN | Cloud Condensation Nuclei |
| 16 | IMD | India Meteorology Department | 39 | TCC | Total Cloud Cover |
| 17 | OC | Organic Carbon | 40 | HCC | High Cloud Cover |
| 18 | AOT | Aerosol Optical Thickness | 41 | MCC | Medium Cloud Cover |
| 19 | MERRA | Modern Era Retrospective-Analysis for Research and Applications | 42 | LCC | Low Cloud Cover |
| 20 | SEDAC | SocioEconomic Data and Applications Centre | 43 | ACF | Autocorrelation Function |
| 21 | ERA | European Re-analysis | 44 | GCM | General Circulation Model |
| 22 | NEO | NASA Earth Observations | 45 | R1/R1A | Region 1/1a |
| 23 | MODIS | Moderate Resolution Imaging Spectroradiometer | 46 | M6/7/8/9 | Month 6/7/8/9 |












| Mon | Threshold | r1 | r2 | r3 | avg |
|---|---|---|---|---|---|
| 6 | 1 | -0.421 | -0.266 | -0.389 | -0.359 |
| 6 | 2 | -0.429 | -0.292 | -0.392 | -0.371 |
| 6 | 3 | -0.433 | -0.298 | -0.396 | -0.376 |
| 7 | 1 | -0.403 | -0.376 | -0.425 | -0.401 |
| 7 | 2 | -0.405 | -0.39 | -0.422 | -0.406 |
| 7 | 3 | -0.406 | -0.397 | -0.421 | -0.408 |
| 8 | 1 | -0.413 | -0.407 | -0.433 | -0.417 |
| 8 | 2 | -0.412 | -0.411 | -0.431 | -0.418 |
| 8 | 3 | -0.414 | -0.411 | -0.436 | -0.42 |
| 9 | 1 | -0.418 | -0.411 | -0.442 | -0.424 |
| 9 | 2 | -0.422 | -0.417 | -0.444 | -0.427 |
| 9 | 3 | -0.424 | -0.419 | -0.449 | -0.431 |

**Table 2.** Selection of thresholds for DDF analysis.

| Region | Case 1(SDP) | | Case 2(MDP) | | Case 3(LDP) | |
|---|---|---|---|---|---|---|
| | Range | Average | Range | Average | Range | Average |
| Region 1a | 8-10 | 9 | 10-14 | 12.5 | 14-18 | 16 |
| Lucknow | 4-12 | 9.5 | 13-17 | 15 | 18-30 | 22 |
| Region 3 | 14-20 | 19 | 21-24 | 22.6 | 24.5-27.5 | 26 |

**Table 3.** Classification of dry day phase according its length.

| Region | Components | | | |
|---|---|---|---|---|
| | Aerosol | SSN | ENSO | SHUM |
| Region 1a | 0.393 | 0.008, | 0.161 | -0.207 |
| Region 3 | 0.107 | 0.078 | 0.056 | -0.267 |

**Table 4.** MLR coefficients for all general factors.

| Region | Components | | | | |
|---|---|---|---|---|---|
| | BC | Dust | OC | Sea Salt | Sulphate |
| Region 1a | 0.542 | 0.129 | 0.263 | 0.326 | 0.124 |
| Lucknow | 0.864 | 0.218 | 0.556 | 0.011 | 0.155 |
| Region 3 | 0.464 | 0.431 | 0.120 | 0.182 | 0.033 |

**Table 5.** MLR coefficients for aerosol components.









| Sl No. | Model Name | Correlation | Normalized STD |
|--------|------------|-------------|----------------|
| 1 | ACCESS 1.3 | 0.20398 | 0.417101 |
| 2 | CAN ESM 2 | 0.3534 | 0.291455 |
| 3 | CMCC CESM | 0.27519 | 0.376355 |
| 4 | CNRM CM5 | 0.51646 | 0.254338 |
| 5 | CSIRO MK 3 | 0.02852 | 0.564645 |
| 6 | GFDL ESM 2M | -0.01922 | 0.649957 |
| 7 | HADGEM2 -CC | -0.23064 | 0.410529 |
| 8 | INMCM4 | -0.05084 | 0.558969 |
| 9 | IPSL CM5 LR | 0.27714 | 0.41382 |
| 10 | MIROC 5 | 0.26838 | 0.362948 |
| 11 | NOR ESM 1M | 0.39618 | 0.283413 |

**Table 6. Performance details of all 11 GCMs used.**

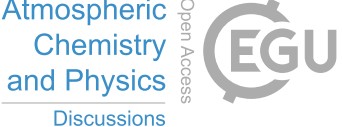

**Figures**

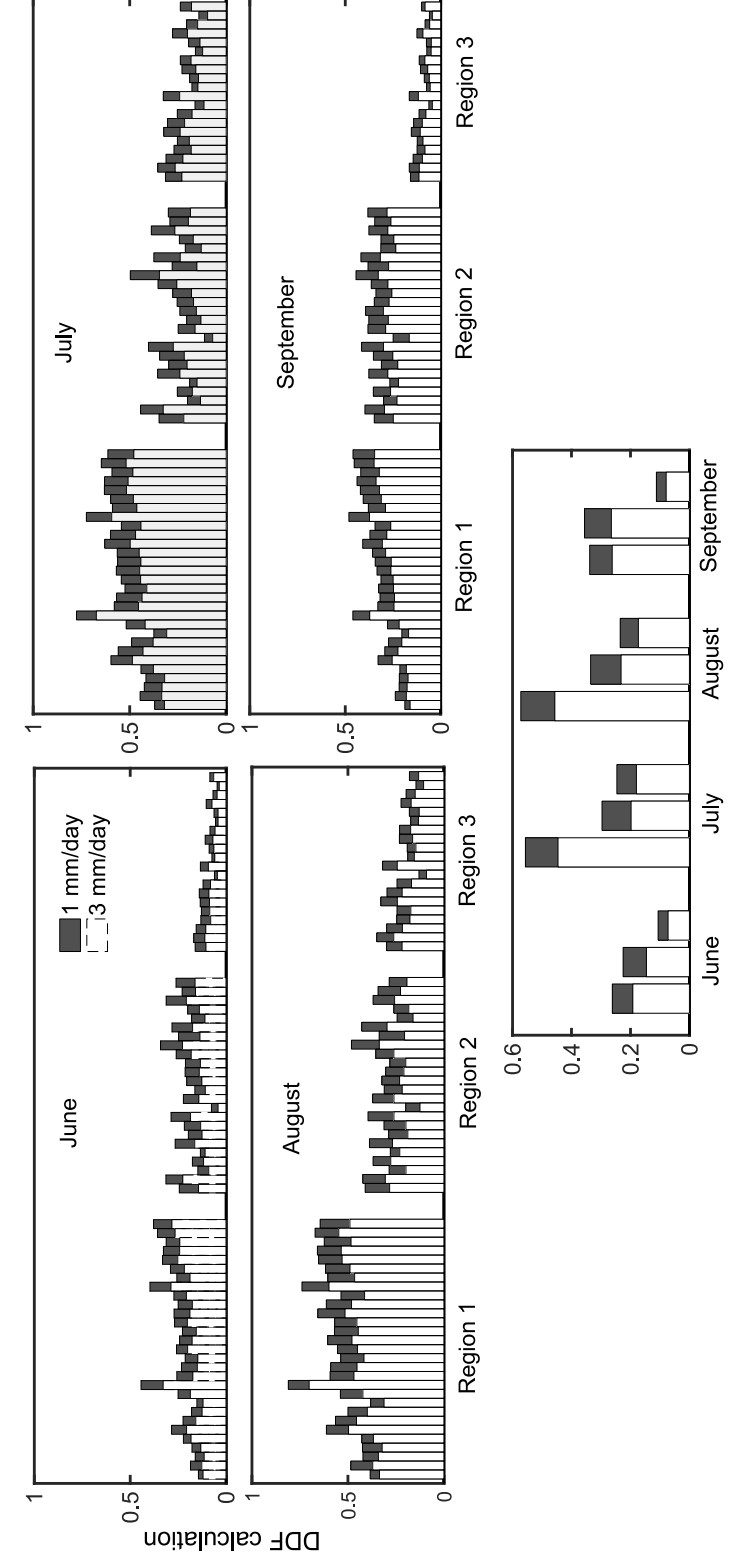

**Figure 1.** Estimation of the optimum threshold for DDF selection using dry day exclusion method.







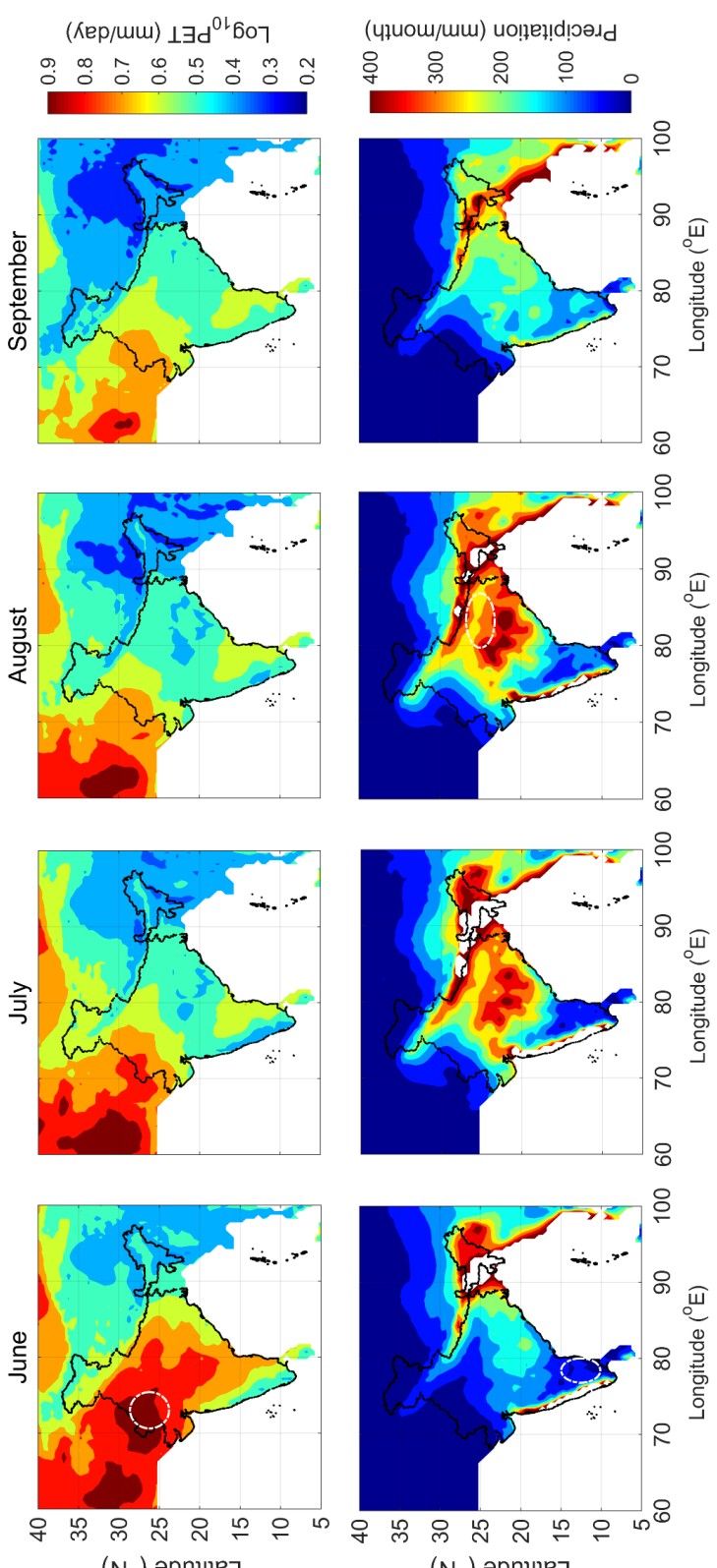

**Figure 2.** Monthly averaged maps of potential evapo-transpiration rate and precipitation during June-September.



10'



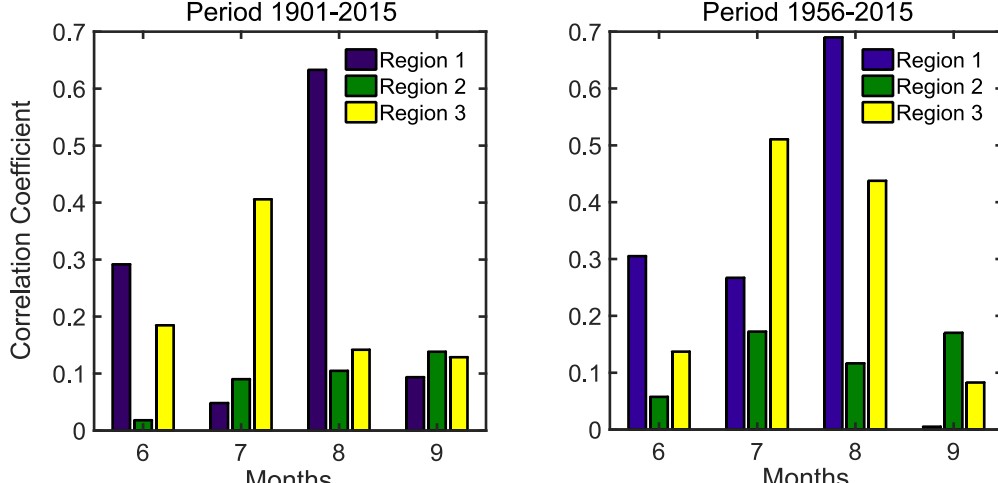

**Figure 3.** Correlation coefficients between DI and DDF values for all monsoon months for two different climatic periods (a) 1901-2015 and (b) 1956-2015.





**Figure 4.** Monthly mean and 15 day trends of DDF for regions 1a, 1b, 2 and 3.









**Figure 5.** Frequency distribution analysis results of various controlling factors behind DDF evolution for various types of dry phase lengths over region 1a, (a) using general parameters like total aerosols, SSN, ENSO and humidity (b) Variation of DDF corresponding to 5 aerosol components such as BC, Dust PM 2.5, OC, Sea Salt and Sulphates.








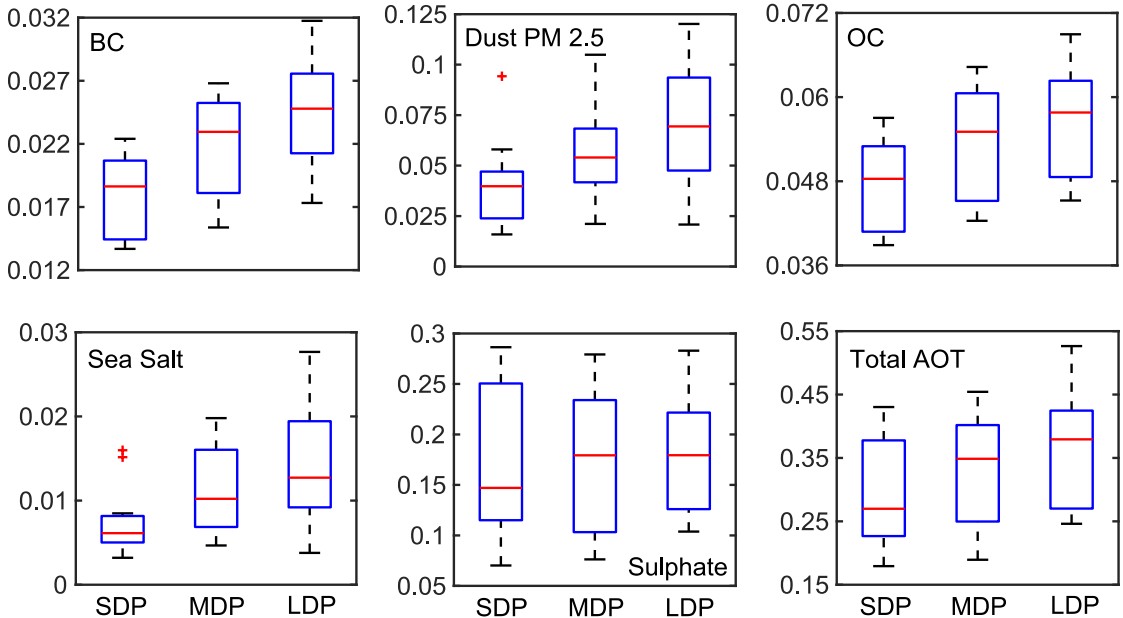

Figure 6. Frequency distribution analysis results of various controlling factors behind DDF evolution for various types of dry phase lengths over Lucknow corresponding to 5 aerosol components such as BC, Dust PM 2.5, OC, Sea Salt and Sulphates.









**Figure 7.** Statistical comparison of the climatology of all parameters during August for region 1, region 1a, Lucknow during various time spans (a) Dry Day Frequency values between 1956-2015, (b) Cloud cover parameters (TCC,HCC,MCC,LCC) during 1980-2015 and (c) Cloud Particle Radius values over 2000-2018





1121

1122

1123

**Figure 8.** Frequency distribution analysis results of various controlling factors behind DDF evolution
for various types of dry phase lengths over region 3, (a) using general parameters like total aerosols,
SSN, ENSO and humidity (b) Variation of DDF corresponding to 5 aerosol components such as BC,
Dust PM 2.5, OC, Sea Salt and Sulphates.

1128





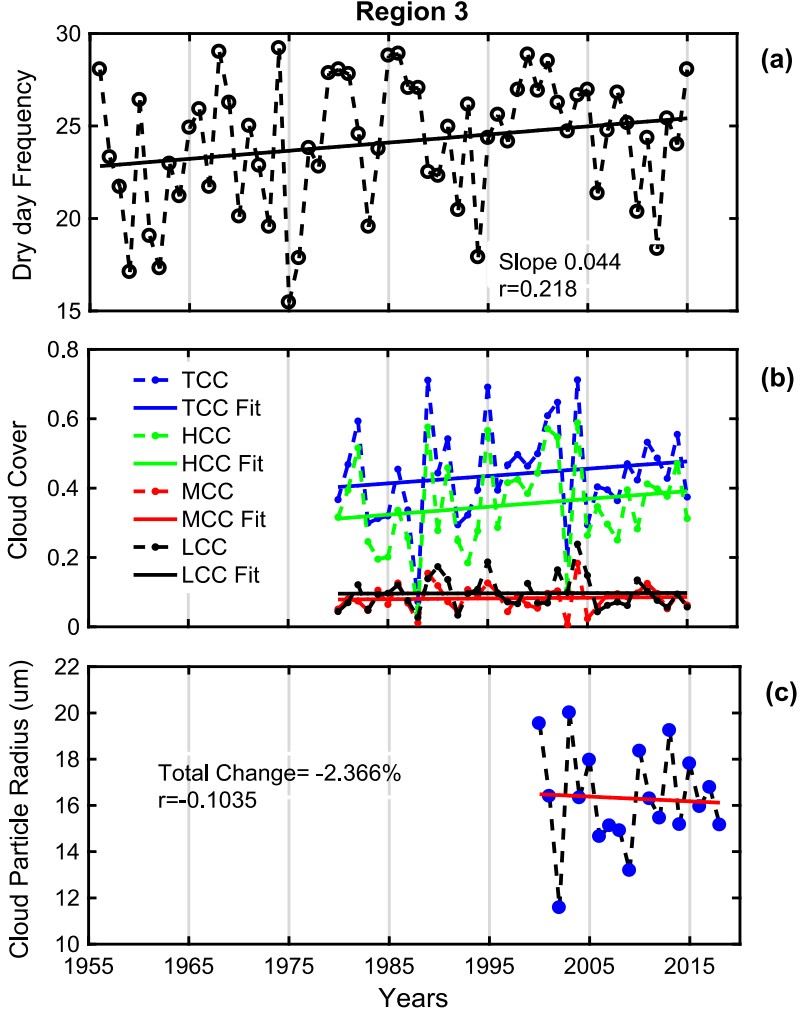

1129

**Figure 9.** Statistical comparison of the climatology of all parameters during July for region 3 during various time spans (a) Dry Day Frequency values between 1956-2015, (b) Cloud cover parameters (TCC,HCC,MCC,LCC) during 1980-2015 and (c) Cloud Particle Radius values over 2000-2018.













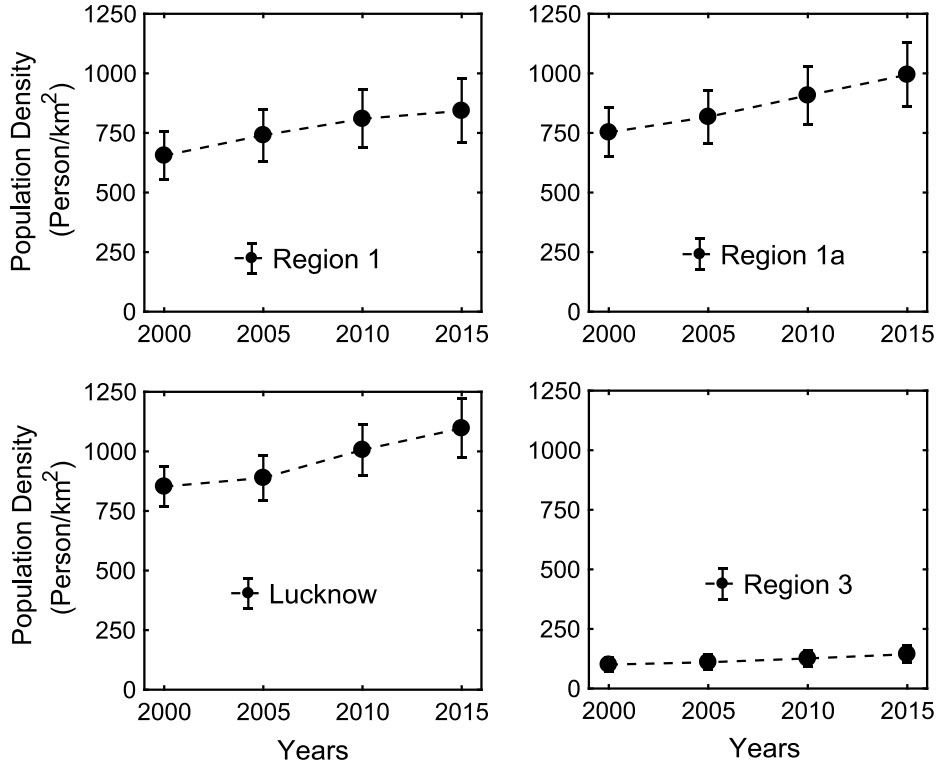


**Figure 10.** Region-wise population densities for Region 1, 1a, Lucknow and Region 3
comprising historical data (2000-2015) and projected data (2020)




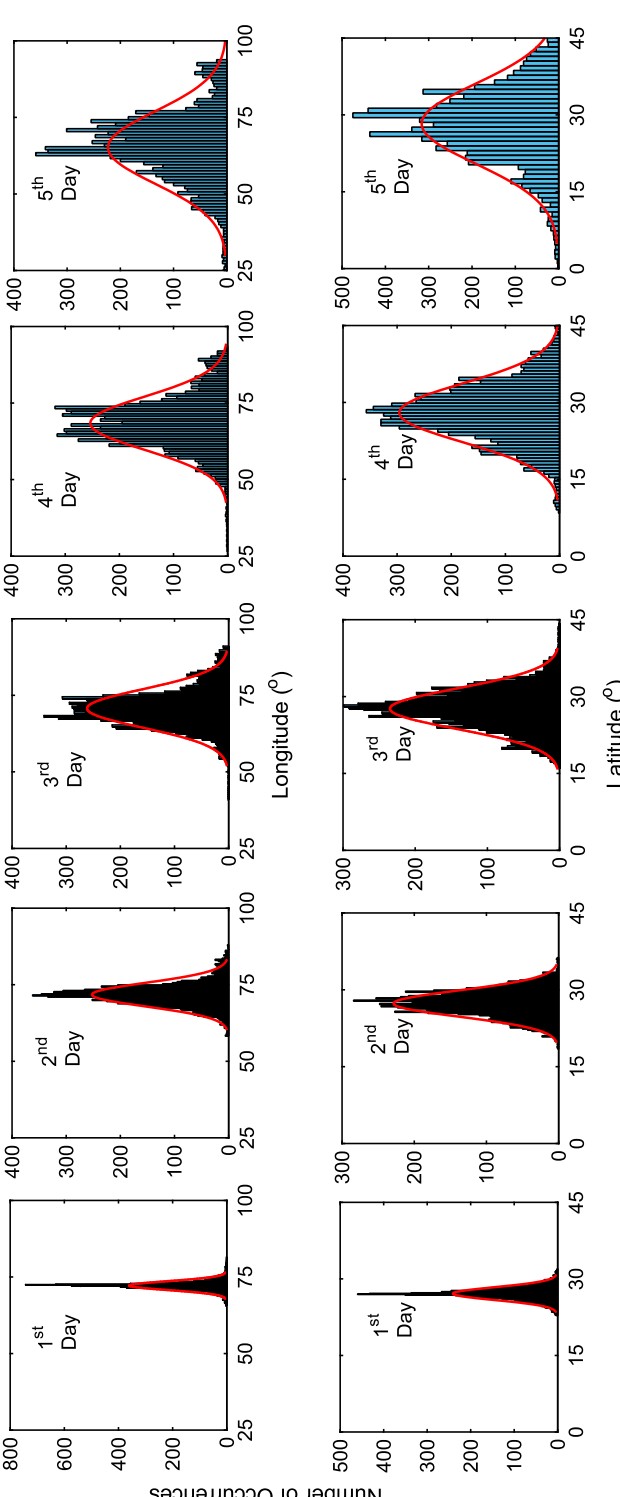

**Figure 11.** Statistical representation of all back trajectory endpoints with respect to latitude and longitude starting from the central grid point (27 N, 72.5 E) of Region 3.







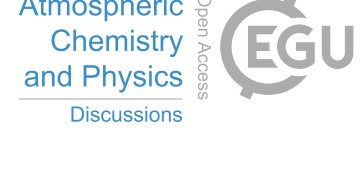



**Figure 12.** (a) Climatic variations in dry day frequency over Region 1 and 3 containing both historical data (upto 2005) and RCP8.5 projections (2006-2100) of multi model mean from 3 selected GCMs (b) Projected lat-lon maps of DDF for all three 50 year periods from 1951-2100.



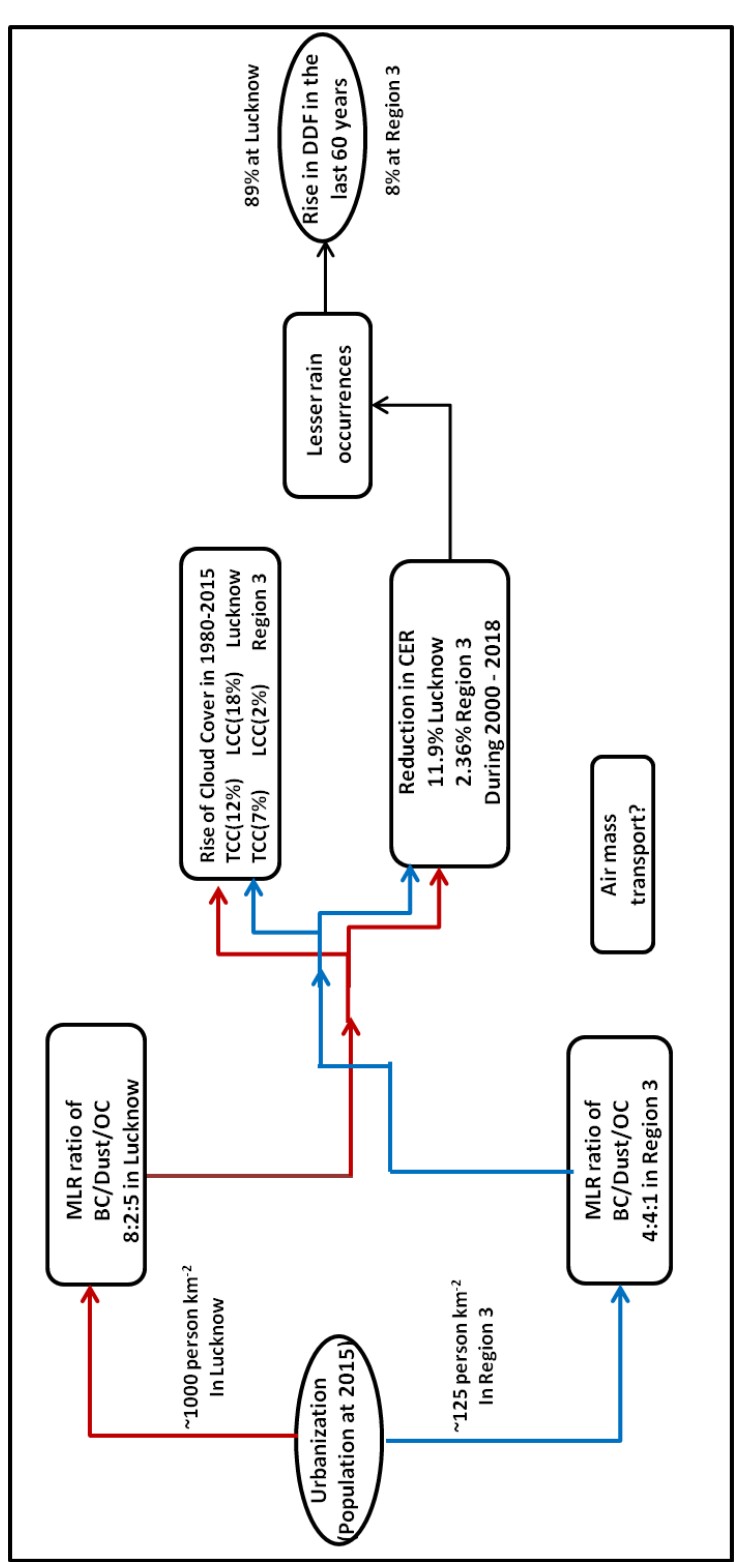

**Figure 13.** Possible mechanism behind the extension of drying phases in Lucknow and region 3 during the mid-monsoon period.

1154
1155
1156