# Peer review of "Growth in mid-monsoon dry phases over Indian region"

_Atmospheric Chemistry and Physics, 2019_

## Referee Comment (RC1) · Anonymous Referee #1 · 4 Mar 2019

The manuscript discusses the influence of anthropogenic aerosols on mid-monsoon dry phases over Indian region.

Major comments -

Data - monthly averaged data of CER have several issues as has been studied by several researchers.

MERRA-2 simulations of aerosol properties over India have not been validated, and they have several issues. These simulations cannot be taken to represent the aerosol characteristics, which is a major input on which the paper relies upon.

[Figure]

Also, as mentioned above while the current aerosol simulations are not validated, how these simulations can be taken to represent the past and future?

Monsoon breaks in Indian monsoon is an important phenomena that contributes to aerosol increase or decrease and contributes to dry days.

The changes in CER are almost nil and statistically insignificant, be it in region 3 or region 1. Also, a few changes are quite abrupt.

Datasets used have different resolution, this is another major issue, because the components exhibit large spatial variations within even 10 km.

Figure 4 - the number of dry days do not show any significant difference between the first and last 60 years except for region 1a.

---

## Referee Comment (RC2) · Anonymous Referee #2 · 17 Mar 2019

The manuscript describes possible correlations between absorbing aerosols and drought over parts of India and Lucknow in particular to increase in aerosol concentrations over the past sixty years and more appropriately for the past 30 years when satellite datasets of aerosol loadings are available. There is a lot of hard work and analysis that went into the study and appreciate the effort put in by the authors. Unfortunately, they are challenged by the lack of specific data sets that will help tease out any such effect and the possibility that this may not be an easy trend to distinguish with the current available data. In the end this manuscript goes in so many directions that it is hard to follow and some of the relationships discussed are thinly supported and probably contradictory. The essential problem is that when there is no rain the aerosol

loading will increase due to the absence of washout. May be this is bit debatable when there is a non-hydrophylic BC is present, but when using AOD as an indicator washout is a fact. Essentially, most of the India with anthropogenic aerosol pollutants that AOD can be expected to recover quickly after a rainfall (washout) event. This implies that the dry days will have more aerosols(AOD) in the atmosphere and there will be a pretty good correlation between dry days and AOD over much of India during monsoon in places with significant anthropogenic pollution. Then to argue that more AOD leads to more dry days, we will need a testable hypothesis. I didn't see any such hypothesis in this manuscript or I might have completely missed it. For example, separating the data into high AOD days of few days and the next rain event, holding all other factors constant, to days with low AOD days with similar set of meteorological conditions and show that the next rainy day is extended by x number of days may prove the authors point. As the manuscript stands now, it doesn't. A second issue I have is the definition of DDF, it described as a frequency. The way it is computed seems like it is simply a daily count of number of days with less than 1mm/day rainfall. When these types of duration statistic is used in hydrology or climate statistics the frequency is given a period 3 day or 5 day (example a heat wave is a 3 day event with greater than 95F maximum temperature in the USA). One day of less than 1mm rainfall does not lead to drought, there has to be a sequence of such days to create this event and this statistic should be modified to prescribe a consecutive number of days over which this is calculated. Third issue is that the DDF and DI as described here are essentially precipitation derived and not independent. It just happens that DDF is high frequency description of DI and one would expect them to be correlated when the precipitation is in the same direction as DDF(low). I don't see the value of correlating DDF to DI in this context. Again, I may be missing something. Fourth big issue is the cloud analysis. It seems like the cloud analysis is used to suggest that there is a decrease in drop size an increase cloud lifetime (more cloudiness) and a decrease in rainfall (drizzle suppression) all lending credence to aerosol second effect. I think this is an interesting insight coming from the data. However, I couldn't figure out where all this data is coming from (MERRA2?).

This needs more supporting data than presented here, I am not even sure what to make of figure 7, so the cloud droplet radius (I am assuming that is what this is? it cloud particle radius for low level clouds?) has decreased by >11% over Lucknow over the past 10 years in MERRA data. Can you also discuss what uncertainties are there is MERRA data estimates of cloud droplet size? This requires a separate paper and much more detailed analysis and support than presented here. Fifth, why sunspots? Is there a physical reason why sunspot activity effects precipitation over India? Last but not least the dust transport analysis seems irrelevant and adds no information to the paper. Also, let us not forget population density thrown into the mix, how does that effect precipitation and dry days and through which physical mechanism?

Overall, this manuscript describes a jumble of statistics trying to connecting what could probably a flawed 'duration' statistic to a number of randomly selected parameters to make a point that is not well supported.

Please also note the supplement to this comment:
https://www.atmos-chem-phys-discuss.net/acp-2019-1/acp-2019-1-RC2-supplement.pdf

---

## Author Comment (AC1) · 26 Apr 2019

First of all we wish to thank the reviewer for his comments/suggestions which significantly improved the content of the manuscript. The authors have addressed all the comments raised by the reviewer and incorporated in the revised manuscript.

Major comments - 1. Data - monthly averaged data of CER have several issues as has been studied by several researchers. Reply: We admit that the monthly averaged variation of CER does not provide any statistically significant trends. This may be attributed to certain factors namely: insufficient time frame (<20 years) for studying the long term variation as well as other reliability issues with the data. On the other hand,

we have shown the evidence of increasing number of dry days even in the presence of increased low cloud cover. Hence this phenomenon indicates the development of second radiative effect of aerosols without the necessity of showing CER trends. However, to improve the contention of this paper, we have removed the CER portion in the manuscript to avoid further ambiguity.

2. MERRA-2 simulations of aerosol properties over India have not been validated, and they have several issues. These simulations cannot be taken to represent the aerosol characteristics, which is a major input on which the paper relies upon. Also, as mentioned above while the current aerosol simulations are not validated, how these simulations can be taken to represent the past and future? Reply: First, we wish to inform that the BC AOT dataset from MERRA 2 is validated against in-situ surface BC concentration measurements over Kolkata, a metropolitan city in Figure S1. Second, there also have been many recent attempts such as Pandey et al. (2017), Randles et al. (2017) and Buchard et al. (2017) which supported the validity of using MERRA datasets for similar climate studies both globally as well as over the Indian region. Finally, we have always used frequency distribution analysis in this paper to show the qualitative relation between aerosols and rainfall. Consequently, we have not focussed on the future trends of aerosols; rather we have exploited their interrelationship with cloud properties to arrive at a possible future trend of rain events.

We have included following references along with discussion in the revised manuscript to support our observations.

Pandey, S.K., Vinoj, V., Landu, K. and Suresh Babu, S., Declining pre-monsoon dust loading over South Asia: Signature of a changing regional climate Scientific Reports, 7: 16062, 2017, DOI:10.1038/s41598-017-16338-w. Randles, C.A., Da Silva, A.M., Buchard, V., Colarco, P.R., Darmenov, A., Govindaraju, R., Smirnov, A., Holben, B., Ferrare, R., Hair, J. and Shinozuka, Y., 2017. The MERRA-2 aerosol reanalysis, 1980 onward. Part I: System description and data assimilation evaluation. Journal of climate, 30(17), 6823-6850, 2017, DOI:10.1175/JCLI-D-16-0609.1 Buchard, V., Randles, C.A.,

Da Silva, A.M., Darmenov, A., Colarco, P.R., Govindaraju, R., Ferrare, R., Hair, J., Beyersdorf, A.J., Ziemba, L.D. and Yu, H., 2017. The MERRA-2 aerosol reanalysis, 1980 onward. Part II: Evaluation and case studies. Journal of Climate, 30(17), 6851-6872, 2017, DOI:10.1175/JCLI-D-16-0613.1

3. Monsoon breaks in Indian monsoon is an important phenomena that contributes to aerosol increase or decrease and contributes to dry days. Reply: Monsoon break phase occurrences may additionally contribute to the increase of dust type aerosols to some extent. However, this investigation is beyond the contention of the present analysis as here we indicate the dominance of anthropogenic aerosols over other natural factors such as ENSO and solar effect in changing the DDF over Indian region.

4. The changes in CER are almost NIL and statistically insignificant, be it in region 3 or region 1. Also, a few changes are quite abrupt. Reply: We have already addressed this issue by keeping away the CER portion in the revised manuscript.

5. Datasets used have different resolution; this is another major issue, because the components exhibit large spatial variations within even 10 km. Reply: It may be noted that we have focussed on three broad study areas each of whom had a minimum aerial coverage of 3x5 Degrees ($\sim$300 – 500 Km on both dimensions) as we want to study the redistribution of monsoon rainfall which is not a local phenomenon. Hence to investigate this qualitative relationship we have used several controlling factors from ERA and MERRA with a resolution of 0.5-0.75 degrees ($\sim$50-80 km). It may be noted that to understand the variability in this meso to synoptic scale aberration of rainfall diversity, a spatial averaging of various factors are computed. Hence, in no possible case, these relationships will change drastically due to mismatched grid resolutions of by 10-20 km.

6. Figure 4 - the number of dry days do not show any significant difference between the first and last 60 years except for region 1a. Reply: We accept that from Figure 4, the number of dry days does not show any significant difference between the first

and last 60 years except for region 1a which has led the full focus of the present study to go towards region 1a. Since region 1b also falls in almost the same zone of upper Indo-Gangetic Plane, hence some changes were also expected there; however such variations are not reflected in Fig 4a. But interestingly, further investigations also showed that the 15 day trend values are notably higher in Region 1a compared to 1b. This is again proved, in later sections where it has been indicated that anthropogenic BC has stronger dominance over DDF in region 1a than in overall region 1. Hence, we have to accept the fact that due to combined effect of all natural and anthropogenic factors, the distribution of dry days did not show that much prominent shift in last 60 years over region 1b as it happened over 1a.

We once again thank the reviewer for providing potential suggestions throughout the paper. This review experience was indispensable in pushing us forward to improve our work.

---

## Author Comment (AC2) · 26 Apr 2019

First of all we wish to thank the reviewer for appreciating actual content of the work and providing constructive comments/suggestions which significantly improved the content of the manuscript. We have addressed all the comments raised by the reviewer and incorporated in the revised manuscript.

1. The essential problem is that when there is no rain the aerosol loading will increase due to the absence of washout. May be this is bit debatable when there is a non-hydrophylic BC is present, but when using AOD as an indicator washout is a fact. Essentially, most of the India with anthropogenic aerosol pollutants that AOD can be

expected to recover quickly after a rainfall (washout) event. This implies that the dry days will have more aerosols (AOD) in the atmosphere and there will be a pretty good correlation between dry days and AOD over much of India during monsoon in places with significant anthropogenic pollution. Then to argue that more AOD leads to more dry days, we will need a testable hypothesis. I didn't see any such hypothesis in this manuscript or I might have completely missed it. For example, separating the data into high AOD days of few days and the next rain event, holding all other factors constant, to days with low AOD days with similar set of meteorological conditions and show that the next rainy day is extended by x number of days may prove the authors point. As the manuscript stands now, it doesn't. Reply: Thanks for nice suggestion. Keeping the reviewer's comment in mind, we tried to investigate the interrelationship between atmospheric aerosols and rainfall distribution for this study region. We already discussed that in the main region of interest (Region 1a and Lucknow), the dry day frequencies are progressively increasing in number during late July- mid September. For instance, we here concentrate on Lucknow which is an urbanized location in Region 1. Next, we show the average meteorological conditions of surface temperature, pressure, winds, moisture content and rainfall accumulation over 16-30 July during 1980-2015 in Figure 1. The long term mean and 2 sigma standard deviations are also shown in the figures to exclude the years having abnormal weather conditions. The diagram shown below suggests that for three years namely: 1980, 1987 and 2002, few met parameters have shown values beyond the general range, hence they are obliterated. In the next attempt, it was required to see the effect of low rainfall periods and AOD on impending DDF for the next few days during these years. Hence a set of years having comparatively lower rainfall accumulation during 16-31 July were identified. A total of 16 years were recorded which had rainfall values between the 50th and 25th percentile of the population. It may be noted that certain years experienced rainfall below the 1st quartile and hence they were neglected to preserve the data uniformity. Next the average AOD values were accumulated for those years and interestingly, two well separated clusters having a set of non-adjacent 8 years in each were observed: one with AOD

below 0.3 and other above 0.4. Now to study the effect of these two AOD clusters on rainfall, their corresponding DDF values are observed for the next 15 days (1-15 August). This time shift was employed in order to investigate the net effect changing AOD on impending rainfall distributions. It was observed that DDF values are distinctly higher for high AOD compared to the lower AOD case. Hence this supports that higher AOD necessarily leads to more DDF in next few days. We have included this new analysis along with discussion in the revised manuscript.

2. A second issue I have is the definition of DDF, it described as a frequency. The way it is computed seems like it is simply a daily count of number of days with less than 1mm/day rainfall. When these types of duration statistic is used in hydrology or climate statistics the frequency is given a period 3 day or 5 day (example a heat wave is a 3 day event with greater than 95F maximum temperature in the USA). One day of less than 1mm rainfall does not lead to drought, there has to be a sequence of such days to create this event and this statistic should be modified to prescribe a consecutive number of days over which this is calculated. Reply: First of all we wish to inform that it is not our intention to indicate that single day instances of rainfall less than 1 mm may lead to a potential drought situation. Instead, we investigate a change in daily rainfall distribution over the mid-monsoon period by studying the total number of dry days over a 15 and 30 day span. Secondly, the month of August experiences very heavy rainfall ($\sim$300 mm) over region 1; hence the number of dry days is expected to be very low (<10) there. However, we have observed that the DDF has become very high in recent years (>20). As a result, it is now possible to create a near-drought situation without the presence of any extended dry phases (3-5 days).This argument is further supported by the presence of a prominent relationship between DDF and drought intensity over the same region. .Nevertheless, to verify the feasibility of the reviewer's comment, we have calculated the length of all continuous dry spells in the month of August which experienced rainfall less than 1 mm. Consequently, the frequency distribution analysis of these continuous dry spell lengths in Figure 2 depicted a very negligible number of cases having length greater than 2 days (as shown in the figure). Now since this data

volume is found to be extremely less for performing further statistical analysis we have continued with the same definition of DDF as used previously in this study.

3. Third issue is that the DDF and DI as described here are essentially precipitation derived and not independent. It just happens that DDF is high frequency description of DI and one would expect them to be correlated when the precipitation is in the same direction as DDF(low). I don't see the value of correlating DDF to DI in this context. Again, I may be missing something. Reply: We wish to clarify that DDF and DI are not controlled by precipitation in similar manner due to a number of reasons. First, DI is dependent on the monthly accumulated difference between precipitations and PET while DDF depends upon the behaviour of daily rainfall accumulation. As the distribution of daily rainfall is not generally symmetric, hence there cannot be any agreement between the day-wise and monthly aggregate data. The second reason in support of the DDF, DI independence is the presence of a third factor called PET which it depends upon a set of fixed (geographical location, season, vegetation and soil type) as well as variable components (temperature (max, min and daily mean), moisture content, wind speed, surface pressure and net radiation flux (which again depends on many other variable components)) but it does not depend on rainfall. As a result of these arguments the correlation analysis between DDF and DI are found significant only in few months and regions and not in all cases as doubted by the reviewer. We have included this analysis along with discussion in the revised manuscript.

4. Fourth big issue is the cloud analysis. It seems like the cloud analysis is used to suggest that there is a decrease in drop size an increase cloud lifetime (more cloudiness) and a decrease in rainfall (drizzle suppression) all lending credence to aerosol second effect. I think this is an interesting insight coming from the data. However, I couldn't figure out where all this data is coming from (MERRA2?). This needs more supporting data than presented here, I am not even sure what to make of figure 7, so the cloud droplet radius (I am assuming that is what this is? It cloud particle radius for low level clouds?) has decreased by >11% over Lucknow over the past 10 years in

MERRA data. Can you also discuss what uncertainties are there is MERRA data estimates of cloud droplet size? This requires a separate paper and much more detailed analysis and support than presented here. Reply: We would like to clarify that cloud lifetime information was derived from cloud cover data as given by the ERA-Interim Reanalysis database while the Cloud Particle Radius was utilized from NASA. Further, we would like to add that the long term trends of cloud particle radius are not found to be statistical significant which actually goes with the question raised by the reviewer (regarding CPR uncertainty). On the other hand, we have shown the evidence of increasing of dry day frequency even in the presence of increased low cloud cover; thus indicating towards the development of second radiative effect of aerosols and this argument negates the necessity of showing CER trends. In this case, cloud radius analysis portion has been removed from the revised version of the manuscript.

5. Fifth, why sunspots? Is there a physical reason why sunspot activity effects precipitation over India? Reply: There have been several scientific mentions underlying the effect of solar intensity on tropical rain and monsoon strengths both over India and abroad in the last few decades. A few references are added according to the reviewer's query namely: Neff, U., Burns, S.J., Mangini, A., Mudelsee, M, Fleitmann, D., and Matter, A., Strong coherence between solar variability and the monsoon in Oman between 9 and 6 kyr ago, Nature 411 (2001) 290 – 293. Agnihotri, R., Dutta, K., Bhushan, R. and Somayajulu, B. L. K., Evidence for solar forcing on the Indian monsoon during the last millennium, Earth and Planetary Science Letters 198 (2002) 521 – 527. We have included these references along with discussion in the revised manuscript.

6. Last but not least dust transport analysis seems irrelevant and adds no information to paper. Reply: Thanks for suggestion. After going through the comment, we realize that the transport analysis is only speaking about additional possibilities which have not been supported in any solid form in the paper. Hence for simplicity, the transport analysis part has also been removed in the revised manuscript.

7. Also, let us not forget population density thrown into the mix, how does that effect precipitation and dry days and through which physical mechanism? Reply: We acknowledge that the correlation between population growth and urbanization (which leads to more BC emission and hence higher DDF) varies depending on several other factors. It may be noted that over India, population rise and over-crowding of cities is a big issue and owing to the dearth of suitable pollution mitigation strategies, the impact of population density is expected to be more dominant over BC emission compared to the other contributing factors such as industrialization and vehicular emissions. To prove this hypothesis, the population density and BC AOT extinction datasets are taken over Region 1, 1a, Lucknow and 3 separately and the variation is shown in Figure 3 shown below. It may be noted that here, the BC AOT values have been averaged during the month of August over a moving window of 5 years to be synchronous with the population density measurements. The figure suggests that population density and anthropogenic aerosols show similar long term variations with the highest values being in Lucknow and lowest in Region3 in both cases. Hence this explains the choice of using population as a proxy of urbanization intensity in this study. We have included the references along with discussion in the revised manuscript. We thank the reviewer once again the reviewer for providing constructive comments/suggestions which made us to improve the manuscript content further.
* * *
[Figure]

Fig 1: Effect of various meteorological parameters such as surface temperature, pressure, PWV, winds and rainfall along with average AOD during 16-31 July with DDF during 1-15 August.

[Figure]

Fig 2: Distribution of extended dry phases

[Figure]

Fig 3: Region-wise population densities and BC AOT values (during August) for Region 1, 1a, Lucknow and Region 3 during 2000-2015, vertical bars represent the corresponding 1 sigma standard deviations values

---

## Author Response (AR2)

**Replies to Co-Editor**

We thank the Co-Editor for his comments/suggestions which significantly improved the content of the manuscript. We have addressed all the comments and have incorporated all corrections in the revised manuscript.

**Major Comments**

**1.** The analysis in the paper does not establish clearly that the increase in the dry day frequency (DDF) is due to aerosols which is the principal claim of the paper. The increase in DDF could also have resulted in increase in aerosols. Thus there could be a mutual dependence of these two factors.

**Reply:** In the previous version of the manuscript, we have done a sequential association analysis where we selected a set of years having almost normal meteorological conditions and rainfall accumulation between 16 – 30 July for Lucknow (an urban case study location). Accordingly, a set of 16 years were obtained which were uniformly divided into two non-overlapping clusters based on their AOD values (during the same period). After that the DDF values for the next 15 days (1 -15 August) were calculated and their cluster mean and variance were plotted with AOD values. The analysis revealed that years having higher AOD in second half of July also experience higher DDF in the first half of August which proves the impact of aerosols on DDF growth. However, the same analysis was not done for other regions in the study. Hence to clear the confusion of the reviewer, the same is also done for Region 1 and 1a along with Lucknow. The analysis has showed almost similar results with slightly higher overlapping conditions for region 1. On the other hand, as the DDF growth of region 3 could not be completely explained by aerosol growth, hence the corresponding cluster analysis in that region has also provided slightly higher overlap. But leaving these minor issues overall, it seems that aerosol growth by itself triggers the dry phase growth. A figure depicting the AOD-DDF relation is shown here (Figure R1) and these figures has also been placed now in the required portions of the revised manuscript as per suggestions from the reviewer.

[Figure]

**Figure R1:** Sequential association between AOD cluster (16-30 July) and DDF (1- 15 August) for Region 1, 1a and Lucknow

We once again would like to thank the editor for providing potential suggestions throughout the paper. This review experience was indispensable in pushing us forward to improve our work.
* * *
**Replies to Reviewer's Comments/Suggestions**

First of all we wish to thank the reviewer for his comments/suggestions which significantly improved the content of the manuscript. We have addressed all the comments raised by the reviewer and incorporated in the revised manuscript.

**Major Comments**

**1.** The primary data on aerosol characteristics (aerosol optical depth (AOD) and contributions to AOD by black carbon (BC), organics, PM2.5 dust, sulphates, and sea salt) are taken from MERRA-2. Several papers published earlier have shown the overall usefulness of MERRA-2 AOD. However, accuracy of the contributions from the individual aerosol components at different geographical regions is not clear, though this product has been compared against observations at a few locations. In order to address this issue, in the present study, the monthly mean near-surface level BC concentration at Kolkata (a polluted region in the Indian subcontinent) has been compared with that of the BC contribution to AOD obtained from MERRA-2 at the same location (Fig.S2). (Here I assume that the contribution of BC to total AOD is the Y-axis of Fig.S2; it is not clear from the text or figure caption). How to validate the columnar contribution of BC to AOD by comparing with the surface values of BC concentration? The inference from Fig.S2 is that the two values are well correlated (which is also expected), but it does not validate the BC contribution to AOD or provide information on the absolute accuracies (bias, slope, uncertainty). Further, the monthly mean values of MERRA-2-derived BC themselves show scatter of +/-0.01 to +/-0.02 (about +/-33% of the mean BC AOD) for the same surface concentration of BC (Fig.S2). Hence, Fig.S2 does not provide information on the absolute accuracies or reliability of the individual aerosol species in MERRA-2. Also, as seen from Fig.S2, the variation of surface BC concentration (varying in the range of 4 to 25 microgram/m3) with columnar BC AOD is linear. What about the non-linear effects due to large variations in effective aerosol single scattering albedo and large changes in multiple scattering contributions for such wide range of BC concentration?

**Reply:** In Fig S2 of the manuscript, the BC component of AOD (shown in Y-axis) has been validated against the surface BC concentrations (in X-axis) over Kolkata. The validation has been progressed with AOD data and not surface concentrations as AOD datasets from MERRA2 have been more explicitly validated with various observational sources over different locations and time frames as pointed out in several previous studies like Randle et al. (2017). We are aware that AOD is calculated as a columnar integral and hence bears information to various phenomena (such as non-linear effects from large variations in effective aerosol single scattering albedo and changes in multiple scattering contributions) at different heights. This possibility is further supported from the deviation in data points from the linear fit line as shown in the figure. However, due to dearth of enough data volume, we cannot verify whether the relation between AOT and surface level concentration is strictly non-linear. We have also sought to access the possibility of using vertical profile-based aerosol extinction/ concentration data. But due to absence of balloon borne measurements or LIDAR observations, this cannot be done either. A dedicated attempt can be taken to examine the absolute accuracy of aerosols from MERRA2 against observational datasets wherever available over India, but it would be out of scope of this study.

On the other hand, we wanted to show the feasibility and not the absolute accuracy of the data which is also supported from the congruence between the frequency distributions of AOT and BC concentrations. Further, we have never talked about any absolute thresholds or values but rather have always talked about certain well separated groups of data after which we have tried to explain the physical mechanisms driving the association (and not direct relationship) between aerosols and mid-monsoon drying.

**2.** It may be noted that MERRA-2 assimilates AOD estimated from MODIS, AVHRR and MISR satellite data and insitu AOD observations from AERONET network. While MERRA-2 globally compares well with the AOD observations (especially over the marine regions), it cannot correct for the deficiencies existing in terms of the missing emissions (e.g., Buchard et al., 2017). More importantly, the satellite-derived AOD used in MERRA-2 was only from AVHRR till 1999, while the subsequent period has seen a major increase in the assimilated satellite data, including MODIS and MISR (see Fig.3 of Buchard et al., 2017). This will enhance the quality of MERRA-2 AOD data during the post-1999 period compared to the period before. The present study focuses on the period between 1980-2015 (Line no. 353). What is the bias or accuracies in MERRA-2 AOD during the periods before and after 1999? This is especially important when the AOD is apportioned into different chemical components.

**Reply:** We accept that satellite-derived AOD of MERRA2 was mainly obtained from AVHRR data till 1999 after which MODIS and MISR is being utilized with ground-based observations from AERONET which has enhanced the quality of MERRA-2 AOD data. In this context, we have found out from MERRA2 Technical Manual by Randle et al. (2017) that the global mean standard deviation between MERRA2 AOD and observations is as small as ±0.013 in case of total AOD and ±0.001 in case of BC and these indicate the feasibility of using the data. However, no such absolute uncertainness values before and after the year 1999 have been quoted in any of the previous attempts especially over the Indian region.

On the other hand, we observed from Randle et al. (2017) that though the magnitudes of AOD and its various components are quite different before and after 1999, yet a prominent overlapping is observed globally between June – October. Since, the present study is also focussed during the month of July-August, hence we have reanalysed the aerosol monthly mean AODs in two clusters of before and after year 1999 respectively. An in-depth statistical analysis of the cluster means show that in most of the cases the cluster mean difference is very small compared to the net variance in the data. Even in case of BC and OC the net rise in mean AOD after 1999 was hardly 60% of the total standard deviation (while it should have been at least >= 150% to be considered significant). This indicates that the mean AOD values remained almost similar over the entire time span of 40 years; thereby addressing the data quality related issues faced in the manuscript. A table showing the extent of overlapping between the aerosol datasets before and after the year 1999 is shown below (Table R2).

| | Region 1a | | | | | | | Lucknow | | | | | | | Region 3 | | | | | |
|---|---|---|---|---|---|---|---|---|---|---|---|---|---|---|---|---|---|---|---|---|
| | BC | Dust | OC | SeaSa | Sulph | TotAer | | BC | Dust | OC | SeaSa | Sulph | TotAer | | BC | Dust | OC | SeaSa | Sulph | TotAer |
| 1980-1997 Mean | 0.015 | 0.032 | 0.038 | 0.015 | 0.143 | 0.265 | | 0.018 | 0.036 | 0.040 | 0.016 | 0.148 | 0.286 | | 0.010 | 0.080 | 0.024 | 0.048 | 0.082 | 0.519 |
| 1998-2015 Mean | 0.023 | 0.036 | 0.051 | 0.018 | 0.203 | 0.372 | | 0.023 | 0.056 | 0.052 | 0.028 | 0.210 | 0.415 | | 0.015 | 0.139 | 0.035 | 0.053 | 0.124 | 0.625 |
| Total Mean | 0.019 | 0.034 | 0.045 | 0.017 | 0.173 | 0.319 | | 0.021 | 0.046 | 0.046 | 0.022 | 0.179 | 0.350 | | 0.013 | 0.110 | 0.029 | 0.051 | 0.103 | 0.572 |
| Difference | 0.008 | 0.004 | 0.013 | 0.003 | 0.060 | 0.107 | | 0.005 | 0.020 | 0.012 | 0.012 | 0.062 | 0.129 | | 0.005 | 0.059 | 0.011 | 0.005 | 0.042 | 0.106 |
| Total STD | 0.013 | 0.061 | 0.023 | 0.029 | 0.231 | 0.199 | | 0.007 | 0.049 | 0.018 | 0.019 | 0.120 | 0.231 | | 0.010 | 0.144 | 0.021 | 0.077 | 0.178 | 0.238 |
| Overlapping | 0.612 | 0.069 | 0.565 | 0.092 | 0.259 | 0.537 | | 0.684 | 0.417 | 0.634 | 0.625 | 0.517 | 0.558 | | 0.433 | 0.409 | 0.550 | 0.064 | 0.234 | 0.356 |

Table R2 The table and related text has been incorporated for the ease of understanding for readers in the supplementary section of the revised version of the manuscript.

**3.** Even if the reliability of the individual chemical compositions from MERRA-2 is acceptable and the comments-1 and 2 given above are ignored, the analysis carried out in the present analysis does not show that the observed increase in dry day frequency (DDF) is caused by aerosols. There is a clear association between the increase in DDF and AOD in several cases (and the individual contributions by some of the species). But this increase in AOD and individual chemical species can (and most probably) be due to increase in dryness, which increases the aerosol production as well as their residence time in the atmosphere, both of which contribute to increased accumulation of aerosols in the atmosphere.

**Reply:** We have already investigated the sequential association between the atmospheric residence of aerosols and development of dry phases over a case study location Lucknow in the previous version of the manuscript. However, to clarify the doubt raised by the reviewer, now this analysis has also been progressed over all the regions mentioned in this study. In this analysis, the first step is to screen out all years during 1980-2015 which depict abnormal meteorological variations. In the next step, a set of years having comparatively lower rainfall accumulation during 16-31 July are identified and the average AOD values of those years produced two very well separated clusters. To study the effect of these two AOD clusters on rainfall, their corresponding DDF values are examined for the next 15 days (1-15 August) over Region 1, 1a and Lucknow. The analysis shows almost similar clustering in DDF with respect to aerosols but the effect of AOD is seen to become more diffused as one shifts from a small urban region Lucknow (having more localised anthropogenic dominance) to Region 1 (having lower urbanization density), which is also well reflected from slightly higher DDF values over Lucknow. After this, the same study is also repeated over Region 3, but in this case the AOD values are observed during 16-30 June while DDF values are taken from 1-15 July. The results from this analysis indicate that the DDF values are much higher over this region due to prevalence of arid climate and not primarily due to aerosols which is also understood from the widespread overlapping

between the two clusters. These explanations have been provided in the revised portion of the manuscript.

**4.** For argument, let us assume that the increase in DDF is due to aerosols, as they can cause changes in clouds, radiation balance of the Earth's surface and atmosphere as well as make atmospheric thermodynamical changes. In Region-1, during the long dry phase (LDP), about 67% of AOD is contributed by sulphates, while the organics, BC, dust and sea salt contribute ~14%, ~6%, ~8% and 5% respectively (this is a rough calculation made from the median values shown in Fig.3). On the contrary, during the short dry phase (SDP), the AOD contribution by sulphates reduces to ~53%, while the organics, BC, dust and sea salt contribute ~21%, ~7%, ~12% and 7% respectively. Overall, the BC contribution remains ~6-7% of the AOD. How does this compare with the aerosol chemical measurements carried out over this region, reported in the literature? The sulphate AOD increased by ~0.15 between the SDP and LDP in Region-1. How such major contribution of sulphates prevails and can contribute to long dry phases? How such a small fraction and weak increase (in terms of magnitude) of BC (with BC AOD of 0.014-0.025) can cause the atmospheric heating or radiation budget changes required for LDP (spanning for 2-3 weeks)? Note that such values of BC (often more, typically 10% by BC mass fraction) prevail over most of the Indian landmass (including southwest India).

**Reply:** In accordance with the reviewer's comment we would like to clarify the following points. First, due to the absence of suitable aerosol particle size distributions measurements and also due to the poor spatial resolution of CALIPSO aerosol profiles, there have been very sparse research attempts which validated the relative contribution of BC to total aerosols especially over the current location. Secondly, it is very difficult to comment on the impact of sulphate aerosols on DDF separately due to inadequate number of previous attempts in this field. Thirdly, an detailed quantitative investigation to understand the specific impact of BC or any other aerosol particles in causing radiative budget changes requires the use of exhaustive in-situ database and strenuous numerical modelling which is not available in this study. Another very important point is that the reviewer has calculated the relative contribution of aerosol components based on the median values and they have made certain % based assumptions. We would humbly like to remind the reviewer that the individual distributions of these aerosol AOTs are not squewed perfectly and they have quite large interquartile ranges and, these component datasets are mutually independent of each other. Hence, it may be possible that yeas having high BC may have either high or low OC or sulphates. In this case the median alone gives a completely biased idea about the % contribution of these aerosols' components over total AOD during the SDP/MDP/LDP period. This is the main reason we have utilized the boxplots rather than showing the mean values with error bars in most of the analysis. A rough idea about the contribution of components can only be authenticated after using intensive observation datasets which is not available in the current scope.

However, in this pretext confusion remains regarding how such small changes in BC have a dominant influence on DDF while Sulphates have relatively no effect on it. As already mentioned before, we do not have real-time observation data or sufficient literature to support this fact theoretically. But, to have a double check on this fact statistically, we have only taken the last 20 years data (as aerosol datasets after 1999 are considered more reliable than the first half) and we have redone the analysis. We have taken all the components of AOD and separated it into two equal clusters of SDP' and LDP'.

Next we have done the boxplot analysis of the AOD values for all the study regions Region 1a, Lucknow and Region 3. Like the previous figure, here also BC and OC has shown clearly increased values in LDP' clusters with very less overlapping. But in the case of Sulphates, despite having larger magnitudes of AOD and showing an increase in mean and median AOD values, it shows a huge overlapping which makes the significance of the cluster separation negligible. Here it may be noted that such overlapping was not observed previously due to presence of MDP which the reviewer may have overlooked in the analysis. To prove that the net increase in Sulphate AOD is negligible with respect to BC, the cluster means of SDP' and LDP' are calculated and the net increase is obtained from their difference. Next the total deviation in the values is calculated and the ratio between the cluster mean increment and the standard deviation (std) are examined. In case of BC the actual growth in cluster mean is ~ 1.45 times of the std while in sulphate it is ~0.5. This implies that though sulphate values show much higher absolute change in LDP but its affect is completely overpowered by the variance or uncertainness in the cluster; but on the other hand, the increase in AOT of BC component is seen with marginal overlapping. This to an extent explains how the sulphate AOD median value increase cannot influence DDF as strong as in the case of BC. A figure depicting the frequency distribution analysis of all aerosol components during the modified SDP' and LDP' classes is shown for reference (Figure R3). This figure will not be added in the revised manuscript to remove additional confusion but a pertinent table in support of this will be provided with some discussion in the appendix section of the manuscript.

[Figure]

**Figure R3:** Frequency distribution of all aerosol components (BC, Dust PM 2.5, OC, Sea salt and Sulphate) for modified SDP' and LDP' classes in Region 1a, Lucknow and Region 3

**5.** Similar scenario prevails over Lucknow (Fig.4). The BC AOD fraction is ~6-7% for SDP and LDP. The increase in BC from SDP to LDP is only ~0.007 (here comes the accuracy of MERRA-2; is this outside the uncertainty limit?), while the AOD increase is ~0.11, of which the contribution of BC is only ~6-7%. Unlike in Region-1a, Lucknow (which is a city located in Region-1a) does not show an appreciable increase in sulphate AOD. Why is this feature distinctly different, when the aerosol residence time can be 3-7 days (during dry phases it can be even more). Observations reported in the literature suggest that the day-to-day variability or variations within a few days in AOD over most of

the Indian region can be significantly more than ~0.1. Further, AODs in the range of 0.3 to 0.8 often prevails over Region-1 and at several other regions in the Indian subcontinent. The median value of AOD prevailing over Lucknow during SDP is ~0.27, which is rather clean. The important question is, can such a relatively small AOD or such small variations (reported here) in AOD or BC AOD cause short or long dry phases? If so, what are the thermodynamical changes or radiative impact produced by such variations? This is not studied in the present work. On the contrary, it is very much possible that dry spells can cause accumulation of aerosols to produce the observed variations.

**Reply:** We would like to bring forward certain important observations. First, the Technical manual of Merra2 by Randle et al. (2017) depicts that the net standard deviation of aerosol components from Merra2 with respect to observation data globally ranges from 0.001 in case of BC to 0.013 in Total AOD. Thus, the issue about small change in BC AOT of 0.007 seems to be addressed. Secondly, here we have talked about a phenomenon of 15 days and 30 days scale, with respect to number of dry days over regions; hence it is important to see the total variability in aerosols rather than the day to day availability. And this mechanism has also been supported in all regions by the sequential effect of aerosols on DDF (with an aerosol residence time of 15 days) as already discussed in previous replies to comments. Third, the reviewer has expressed a concern that if the AOD values are less over SDP then it should not bear any signature towards DDF. We accept this comment, but we want to clarify that the SDP cluster is only shown to draw a contrast with LDP and certain MDP cases where atmospheric aerosols can have some impact on DDF (note that AOD values during LDP are almost double of SDP). Fourth, we have already supported our claim statistically in the previous reply where we showed that BC values exhibit a distinct change with lesser overlapping while the uncertainties in cluster formation in case of sulphates clearly indicate that they do not have any deterministic effect on DDF. Finally, we agree that we have not considered the thermodynamic or circulation-based aspect in previous versions of the manuscript, but now we have added them also in DDF sensitivity analysis which is described in detail in later replies.

**6.** Long-term trends in cloud occurrences shown here are very interesting. However, most of the aerosols being limited to the lower atmosphere, the increase in AOD (which is proposed to have produced the dry spells) should have first affected the low level clouds. In contrast, Fig.5 does not show any increase in low level clouds in Region-1 or 1a, but produced significant increase in high level and total clouds. However, the low level clouds did show a weak increase over Lucknow. Why only at Lucknow? Overall, the increase in middle, high and total clouds is much larger than that in the low level clouds. How does this happen? Is it possible that the increase in high level clouds and consequent increase in greenhouse warming has also contributed to the dryness occurrence? (The variations between low level clouds and aerosols itself can be an interesting study).

**Reply:** Thanks for the comment and appreciation. We wanted to show the impact of aerosols on all types of cloud cover in the previous version of the manuscript. However it is a fact that aerosols are mainly limited to the lower atmosphere; hence the increase in AOD should have mainly affected the low-level clouds. Consequently, this time we have only considered the low cloud cover for analysis. Further, we do not have much idea about which the probable mechanism leading to this increase in high and total cloud cover over the mentioned regions; hence we have been removed in the revised manuscript. After readjusting the scales of low cloud cover data to obtain best fit for region 1, 1a and Lucknow, we have found an increase of cloudiness magnitude in almost all three cases. But then

again, the growth in LCC is most prominent over Lucknow. The reason for this is the fact that Lucknow and its surrounding regions have higher anthropogenic emissions due to more urbanization background which has depicted a prominent impact on DDF growth compared to other regions of more spatial extent like region 1, 1a where the anthropogenic emission dominance on cloud processes is expected to be diffused due to many other factors. This hypothesis can be explained much better by looking at Figure 8 of the old manuscript.

A brief discussion of low cloud cover and deletion of unwanted text is now been done in the revised manuscript in accordance with the reviewer's comment.

**7.** In Region-3, AOD is quite high (median values ~0.55) and comparable for SDP, MDP and LDP. However, based on the average or median values, the total contribution from the individual species (BC, dust, OC, sea salt, sulphate) contributes only ~60% of the total AOD. What are the other components that contribute ~40%? The increase in BC AOD (median) between SDP and LDP is ~0.004 while that of dust AOD is ~0.04. In contrast, sulphate (and OC by a negligible magnitude) has decreased by ~0.03 between SDP and LDP. Why is this contrasting behaviour and how does it compare with the Region-1 (though the precipitation mechanisms at these two places are different, why the role of aerosols is contrasting?).

**Reply:** In general, all the five aerosol components given by Merra2 (BC, Dust PM 2.5, OC, Sea Salt and Sulphates) account for >90% of total AOD in average over Region 1a. But in case of region 3 these parameters account for only ~77% on an average. This large deficit is only prevalent over Region 3 (experiencing arid climate) due to the omission of dust AOT (not Dust PM 2.5). These particles were already represented by the Dust PM 2.5 values in the manuscript hence were obliterated to remove further confusion. However, after incorporation of Dust component, the sum of all AODs now account for more than 95% of the total AOT (not shown here as it is out of the present scope).

Though OC does not show much of this decreasing nature, but the median values of Sulphates have really shown a decrease in the frequency distribution plot over Region 3 only. This has come as an exception to the entire analysis and we do not have any pertinent explanation for it. Moreover, this exception does not affect the main contention of the study, hence may be neglected.

**8.** I presume that the data on aerosol characteristics presented in Figs. 3, 4 and 6 are for the respective days of different dry phases (SDP, MDP and LDP). On the contrary, the analysis on AOD versus DDF shown in Fig.S7 (which is interesting) considers AOD during 16-31 July and DDF during 1-15 August. This may have a basis: aerosols may cause the atmospheric thermodynamical and circulation changes, which may result in dry spells subsequently. In that case, why the analysis shown in Figs.3,4, and 6 used simultaneous data for AOD and DDF?

**Reply:** We have shown the overall association between DDF and aerosol concentrations in a simultaneous time scale for region 1a, Lucknow and Region 3 in the main figures only to depict the relative dominance of anthropogenic aerosols over all other components. However to additionally explain the sequential impact of aerosols on DDF we have made the sequential AOD- DDF clustering analysis for all the regions in the study and we have now added these figures just below those main figure in revised manuscript to remove any confusion.

**9.** In summary, this study shows that there is an association between the increase in dry day frequency and AOD (including some of the individual species) in some regions. As stated earlier, the increase in AOD can be a result of increased dry day frequency as well. This paper does not provide any evidence to show that the increase in dry day frequency is caused by aerosols. Even if it is so, the important question remaining is whether the observed magnitude of increase in aerosols (and individual components) is sufficient to produce SDP, LDP, etc.

**Reply:** This has already been addressed by us using sequential analysis and also by analysing AOD contribution overlapping over all regions using better quality-controlled aerosol data after 1999.

**10.** On the contrary, the manuscript does not present the role of changes in atmospheric circulation pattern, atmospheric thermodynamics, radiation balance or surface temperature variations among SDP, MDP and LDP, all of which are expected to be important (all of which can be also produced by aerosols and other climate forcing mechanisms) in producing dryness and increase in its occurrence (Example, Raman and Rao (1981) on the relationship between blocking highs and droughts; Krishnamurti et al. 2010, etc).

**Reply:** As already mentioned before, an investigation of radiation balance over these regions cannot be done due to dearth of in situ aerosol profiles. But apart from it, as the reviewer suggested, now meteorological parameters such as surface temperature, thermodynamics such as instability parameters (namely vertical index) and circulation patterns such as 850 hPa geopotential height has been added into all the three analysis levels like Distribution analysis, PCA and multi-linear regression test. The analysis of these parameters is not done over Lucknow as it falls inside Region 1a. The investigation reveals that surface temperature and atmospheric instability exhibit an increase (~ 0.5 K) from SDP to LDP in Region 1a but it cannot be considered significant due to prominent overlapping among the clusters. However, this association becomes more diffused in Region 3 and thus it follows that these parameters do not have much impact on dry phases in various regions. A sample of the meteorological parameter variation for region 1 and 3 is shown with total AOT for reference. The same is also supported from the PCA and MLR analysis in Figure R4 thereby addressing the comment.

[Figure]

**Figure R4:** Yearly variation of Surface temperature, Geopotential height at 850 hPa, Vertical Totals Instability index with total AOT for various classes of dry phase length during August for region 1nd July for Region 3.

The updated figures, tables and text have been readjusted in the manuscript wherever needed.

**11.** Overall, the manuscript (text) is too lengthy for conveying the message presented in it. At the same time, some of the very important information are not provided: e.g., the individual species contributions (like BC, dust PM2.5, OC, etc) are the contributions to AOD, proper, incomplete Fig/Table captions (e.g., Fig.S1, S2,S4, S11 (even axes titles are missing), Table-S2, Table-S3, Tables 1,2 and 3). Readability of the manuscript has to be significantly improved.

**Reply:** In the previous review, one of the reviewers have particularly quoted that the manuscript length suits the ACP standards and that it is able to explain its statistical association properly. Species contribution of the aerosol components to AOD is also not done due to its random year to year variability of each component which is mutually independent of each other. The overlapping natures of the clusters have already been explained in replies to previous comments which explain why cluster analysis is used instead of absolute value-based thresholding in majority of this manuscript. Finally, we have done a double check to add axes title and figure description wherever applicable.

**Other comments**

**1.** Table-1: Why is the range of dry days for short dry phase (SDP) different for different regions? Same is the case with MDP and LDP.

**Reply:** It may be noted that Region 1 and 3 fall under two different climatic regimes. Region 3 experiences arid climate as a result of which the number of dry days is almost double of Region 1a and Lucknow in case of SDP and MDP. Keeping a fixed threshold of DDF would have created a biased picture, namely: Region 1 would have always shown SDP and MDP while Region 3 would have always shown LDP. To preserve the relevance of boxplots, the size of all three clusters were required to be kept almost equal, hence this uneven classification.

**2.** In Fig.2(a), correlation coefficients between DI and DDF are positive. Then, why there is a decreasing trend in Fig.S4 (R1, M9; R2, M9; R3, M9; R1, M8; R3, M7, etc)?

**Reply:** In this analysis, long dry period (LDP) conditions have more probability to create droughts with negative DI. Hence, DDF and DI are inversely proportional to each other. We forgot to mention that the absolute values of the correlation coefficients have been plotted in Figure 2. This has now been added in the relevant text of the revised manuscript.

**3.** Lines 245-250: What is the validity of this assumption when the region experiences intra seasonal oscillations and active and break spells?

**Reply:** This study investigates the formation of prolonged dry phases which can also be considered as a type of break spells in monsoon. Hence in these cases the rainfall would be marginal while PET would increase. On the other hand, the impact of prolonged wet spells or active phases in this study has been obliterated as we have taken only those years for sequential analysis where the total rainfall accumulation was between 25-50 percentile of the total rainfall climatology (to maintain uniformity in the hypothesis).

**4.** Fig.S4: DI has both positive and negative values even when the number of dry days in a month is 29-30 (e.g., R3, M6; R3, M8; R1,M6). What could be the mechanism?

**Reply:** It may be noted that the temporal extent of the study was limited to August (M8) for Region 1 and July (M7) for Region 3 for which such exceptions were not observed. However, Region 3 has shown a cluster of few years where DI was abnormally positive even in the presence of extremely high DDF. As already shown in the manuscript, these anomalies may be due to the dominance of natural aerosols (Dust) over anthropogenic aerosols (BC, OC) which could not be explained in the study. On the other hand, certain exceptions were also seen in Region 1, but number of such cases is relatively small. The physical mechanisms behind the occurrence of such anomalies cannot be approximated at present without a separate analysis which does not fit with the current scope of the study.

**5.** Lines 412-414: Regarding sulphate AOD - This is incorrect and against what is seen in Fig.3. It showed a distinct increase from 0.1 to 0.25 between SDP and LDP.

**Reply:** We accept that it is a mistake in explaining the figure. The sulphates were not taken in to analysis in spite of a prominent increase owing to the widespread overlapping between MDP and LDP. This correction has been done in the text of the revised manuscript.

**6.** Lines 450-452: "The distribution analysis on total aerosol AOT shows much larger values over Lucknow than in region 1a and also the variability of the median values with the quartiles and whiskers are also far more deterministic here …". This statement is incorrect as is evident in the AOD variations shown in Figs. 3 and 4. In fact, distinct increase in AOD (median and distribution) between SDP and LDP is better seen in Fig.3 (Region 1a).

**Reply:** We partly disagree with the reviewer. The median and upper quartile values are much higher in Lucknow than Region 1a. But as per as the distinctness of the distribution is concerned, Region 1a looks slightly better especially between MDP and LDP. This line has now been suitably modified in the revised manuscript.

**7.** Line-470: "hence the dependence of dry days can be primarily associated with urbanization". This statement is not supported by the facts at this stage. Or refer to Fig.8 here while making this statement.

**Reply:** This line has now been suitably modified by adding reference to Fig.8 in the revised manuscript.

**8.** Lines 503-504: "Figure 5 reveals that region 1 has a weak but discernible increase from 5 to 15 days in last 60 years". Is this correct? The mean trend line shows an increase from 9 to 13 days only. Similarly, the trend line for Lucknow shows an increase from 9 to 17 (and not 4 to 25 days as given in Line 511). The mean long-term trend over the 60 years and the scatter in the values (year to year variability) should be stated unambiguously.

**Reply:** The authors accept the suggestion of the reviewer. These changes have now been done so that the actual trend fit and not the observation points are presented to clear the confusion of the readers.

**9.** Line 529: "… reduction in cloud particle size …". This is not shown in the manuscript.

**Reply:** This line has now been suitably modified in the revised manuscript.

10. Lines 548-550: "Another periodicity is expected to lie at ~1-2 years which represents the year-year varying component of urbanization." What is the year-to-year varying component of urbanization; how urbanization can have a periodicity of 1-2 years?

**Reply:** We accept that it is a typographic error presented in the manuscript. Actually, a primary peak (of 1-year periodicity) is observed in addition to the 4-year ENSO periodicity due to the presence of various anthropogenic elements such as BC and OC in the ACF analysis. These lines have now been rewritten to remove the confusion of the readers.

**11.** Unnecessary words may be avoided (words like 'now', 'next'', etc are used unnecessarily at several places; Also see usages like 'methodically introduce' (line-13), 'crucial concern' (line 28), 'third and final' (Line-171), 'considerable conditions' (line-225), 'whether daily' (line-257), 'a set of various components' (line 275), 'over mentioned regions'(subtitle, Line 290), 'all throughout in 1a' (line -335), 'which demand primary importance throughout the study' (lines-339-340), 'ENSO oscillations' (line 355), 'previous attempts taken' (line-357), 'the distribution of total aerosols start increasing' (Lines 368-369), 'rain cloud' (Line 399; avoid 'rain' here, as the cloud burning effect due to aerosol absorption is possible for non-precipitating clouds as well), 'an attempt has been progressed over Lucknow… ' (Line 477), 'slant rise in dry days' (Line 621), 'rlilpl'(Line 675), 'A series of investigations are progressed which infer …"(Lines 722-723). This is not a complete list.

**Reply:** All these issues and similar ones have now been addressed in the revised manuscript.

**12.** Missing reference: Tyalagadi et al. 2015.

**Reply:** This reference is now added in the revised manuscript.

**13.** Lines 329-331: What does this mean?

**Reply:** We primarily meant to say that the DDF trends are very weak (< 5%) in most of the 15-day time slots for Region 3. However, the month of July has shown a weak increase in DDF (~10%) which is probably due to alternating precipitation phases in June and August. On the other hand, the DDF trends are much stronger over Region 1. Hence this region will be examined with more emphasis throughout the majority of the study compared to region 3 which is being given secondary importance. These lines have now been simplified and inserted into the revised version of the manuscript.

**14.** Line 430: Modify as : "… 0.542, 0.129, … and 0.124 for BC, dust … and sulphates respectively". Similar is the case in all places where MLR coefficients are given in this manuscript.

**Reply:** These changes have now been done in the revised manuscript.

We once again thank the reviewer for providing potential suggestions throughout the paper. This review experience was indispensable in pushing us forward to improve our work.